# Imaging of pH *in vivo* using hyperpolarized $^{13}$C-labelled zymonic acid

Stephan Düwel[1,2,3], Christian Hundshammer[1,2], Malte Gersch[2,†], Benedikt Feuerecker[1], Katja Steiger[4], Achim Buck[5], Axel Walch[5], Axel Haase[3], Steffen J. Glaser[2], Markus Schwaiger[1] & Franz Schilling[1,2,3]

Natural pH regulatory mechanisms can be overruled during several pathologies such as cancer, inflammation and ischaemia, leading to local pH changes in the human body. Here we demonstrate that $^{13}$C-labelled zymonic acid (ZA) can be used as hyperpolarized magnetic resonance pH imaging sensor. ZA is synthesized from [1-$^{13}$C]pyruvic acid and its $^{13}$C resonance frequencies shift up to 3.0 p.p.m. per pH unit in the physiological pH range. The long lifetime of the hyperpolarized signal enhancement enables monitoring of pH, independent of concentration, temperature, ionic strength and protein concentration. We show *in vivo* pH maps within rat kidneys and subcutaneously inoculated tumours derived from a mammary adenocarcinoma cell line and characterize ZA as non-toxic compound predominantly present in the extracellular space. We suggest that ZA represents a reliable and non-invasive extracellular imaging sensor to localize and quantify pH, with the potential to improve understanding, diagnosis and therapy of diseases characterized by aberrant acid-base balance.

[1] Department of Nuclear Medicine, Klinikum rechts der Isar, Technical University of Munich, Ismaninger Str. 22, 81675 Munich, Germany. [2] Department of Chemistry, Technical University of Munich, Lichtenbergstr. 4, 85748 Garching, Germany. [3] Institute of Medical Engineering, Technical University of Munich, Boltzmannstr. 11, 85748 Garching, Germany. [4] Institute of Pathology, Technical University of Munich, Trogerstr. 18, 81675 Munich, Germany. [5] Research Unit Analytical Pathology, Helmholtz Zentrum München, Ingolstädter Landstr. 1, 85764 Neuherberg, Germany. † Present address: MRC Laboratory of Molecular Biology, Francis Crick Ave, Cambridge CB2 0QH, UK. Correspondence and requests for materials should be addressed to F.S. (email: fschilling@tum.de).

Maintaining acid-base balance is critical for the survival of living species since cellular processes are highly sensitive to changes in proton concentrations. In humans, pH is mainly regulated by the $CO_2/HCO_3^-$ buffer within a narrow pH range, in the blood between pH 7.35 and 7.45. Locally, deviations from the systemic pH are often caused by pathologies, such as cancer, inflammation, infection, ischaemia, renal failure or pulmonary disease[1]. Tumours can acidify their extracellular environment during aerobic glycolysis and increased export of lactic acid, an effect that can even be further enhanced by the reduced buffer capacity of tumour interstitial fluid and poor tumour perfusion[2]. Because of its potentially broad impact, non-invasive imaging of local pH changes has been a major goal in biomedical research, even though so far no technique to measure extracellular pH has been applicable in the clinic.

The majority of approaches for non-invasive pH imaging focused on magnetic resonance imaging (MRI) because of its high spatial resolution and excellent soft-tissue contrast. In addition, optical methods[3] and radioactive tracers[4] for positron-emission-tomography (PET) have been developed. Magnetic resonance spectroscopy (MRS)-based pH sensor molecules exploit pH-induced changes in nuclear magnetic resonance (NMR) parameters, such as chemical shifts of individual atomic nuclei, which are sensitive to the protonation state of the molecule and therefore the surrounding pH[1]. To increase sensitivity, gadolinium and lanthanide complexes as well as iodinated contrast agents with pH-dependent chemical exchange or relaxation properties were developed[5–7]. Endogenous amide proton transfer chemical exchange saturation transfer experiments[8] utilizing the pH-dependent proton exchange from predominantly intracellular proteins are currently used to study pH in the brain in human.

Dissolution dynamic nuclear polarization (DNP) revolutionized MRS by lifting nuclear spin polarization to a so-called hyperpolarized state leading to a sensitivity gain by more than four orders of magnitude[9]. Most prominently, hyperpolarized [1-$^{13}$C]pyruvic acid is currently being used in clinical studies to examine its use for metabolic imaging of prostate carcinoma[10]. Hyperpolarized $^{13}$C-labelled bicarbonate has been proposed as a probe for clinical pH imaging based on the determination of the ratio of $CO_2/HCO_3^-$ and recently, spectroscopic pH imaging methods have been revived and demonstrated in vitro including hyperpolarized $^{15}$N-pyridine derivatives[11] and hyperpolarized $^{13}$C-labelled Good's buffers[12]. Still, a non-invasive clinical method for pH imaging is lacking so far.

In this work, we introduce hyperpolarized [1,5-$^{13}$C$_2$]zymonic acid (ZA) as a novel probe for MRI of pH. We demonstrate that using ZA we can non-invasively image extracellular pH both in vitro and in vivo in the bladder, the kidneys and a tumour model. ZA's non-toxicity, its long lifetime of the hyperpolarization enhancement and its strong sensitivity to pH changes render this new technique valuable for further preclinical and clinical studies using extracellular pH as an imaging biomarker to characterize pathologies with aberrant acid-base balance.

## Results

**Synthesis and hyperpolarization of ZA.** While studying metabolism in tumour cell spheroids using hyperpolarized pyruvate for $^{13}$C NMR spectroscopy[13], we noticed two additional unassigned peaks, whose changes in chemical shift correlated with the extracellular pH in the cell medium (Supplementary Fig. 1). We found unassigned peaks with similar chemical shifts in the literature[14–16]. NMR spectra with unlabelled and fully $^{13}$C-labelled, highly concentrated pyruvic acid revealed the presence of three chemical impurities of pyruvic acid, and we identified the pH-sensitive molecule as ZA, based on its $^1$H

and $^{13}$C chemical shifts and coupling constants (Supplementary Figs 2 and 3)[17].

When synthesized[18] (see Methods for details) from selectively $^{13}$C-labelled [1-$^{13}$C]pyruvic acid, ZA bears two $^{13}$C NMR-active spin ½ nuclei and is suitable for long-term storage. In aqueous solution, ZA slowly hydrolyses to parapyruvate (PP) with a half-life of ~2.5 h (Supplementary Fig. 4). ZA thus stays chemically intact long enough for experimentation. To increase sensitivity in $^{13}$C experiments, we developed a hyperpolarization protocol for ZA, and used simultaneously polarized, metabolically inactive $^{13}$C urea[19] as a chemical shift reference, although a pH determination based solely on the difference in chemical shift of the two $^{13}$C nuclei of ZA is possible as well, but with a lower dynamical range (Supplementary Fig. 5). This resulted in a solution polarization level for ZA of $22 \pm 2\%$ ($n = 3$, see Methods section) with relatively long longitudinal relaxation times ($T_1$) at 3 T for both ZA$_1$ ($T_1 = 43 \pm 3$ s, $n = 3$) and ZA$_5$ ($T_1 = 51 \pm 4$ s, $n = 3$) in 80 mM Tris buffer in $H_2O$ (Supplementary Fig. 6). For back-calculation of pH from changes in chemical shift of ZA, we determined a calibration curve by measuring the hyperpolarized chemical shifts of both ZA$_1$, ZA$_5$ and urea for pH values covering the physiologically and pathologically relevant pH range (Fig. 1). The determined p$K_{a2} = 6.90$ from the hyperpolarized $^{13}$C NMR data agrees well with data from literature[20].

***In vitro* pH measurement.** We then performed in vitro pH imaging measurements in buffer and blood phantoms. Due to its robustness towards gradient errors, motion and flow artefacts, as well as off-resonance effects, we used a free induction decay chemical shift imaging (FIDCSI) sequence leading to a full spectrum for each voxel[21]. Back-calculation of pH on a voxel-by-voxel basis from the acquired spectra results in a spatially localized pH map for a buffer (Fig. 2a) and a blood phantom (Fig. 2b) of compartments pre-adjusted to different pH values. Experiments with varying buffer compositions and conditions demonstrated the sensor's robustness. A linear fit shows excellent correlation ($R^2 = 0.99$) between the $^{13}$C biosensor pH values determined from hyperpolarized buffer and blood phantom experiments and subsequent electrode pH measurements (Fig. 2c). Notably, ZA's pH sensor ability is not influenced by the addition of calcium ions.

***In vivo* probe characterization.** The toxicity of injected probe molecules often limits their use in vivo. HeLa cells showed no reduction in cell viability when exposed to ZA concentrations relevant during an in vivo experiment. Non-toxicity was further substantiated by dose escalation experiments in three rats (Supplementary Figs 7 and 8) and a toxicopathological study (Methods section, Supplementary Table 1 and Supplementary Figs 9 and 10).

Since concentration of ZA and local temperature could vary substantially in vivo upon injection, the influence of these two parameters on a pH determination needs to be known for a robust measurement. We thus recorded $^{13}$C chemical shifts of ZA and urea as a function of pH at three different concentrations of ZA. No differences between the three series of measurements could be detected, making ZA a concentration-independent pH sensor within the experimentally relevant concentration range (Supplementary Fig. 11a). We then tested for a temperature dependence of the relevant p$K_{a2}$ by regular acid-base titration of ZA solutions at different temperatures. Diprotic acid titration curves revealed a very weak temperature dependence of $\Delta pK_{a2}/\Delta T = -0.015 \pm 0.005$ pH per °C (Supplementary Fig. 11b). Hence, ZA's pH sensitivity is a concentration and temperature-independent effect within biologically relevant conditions.

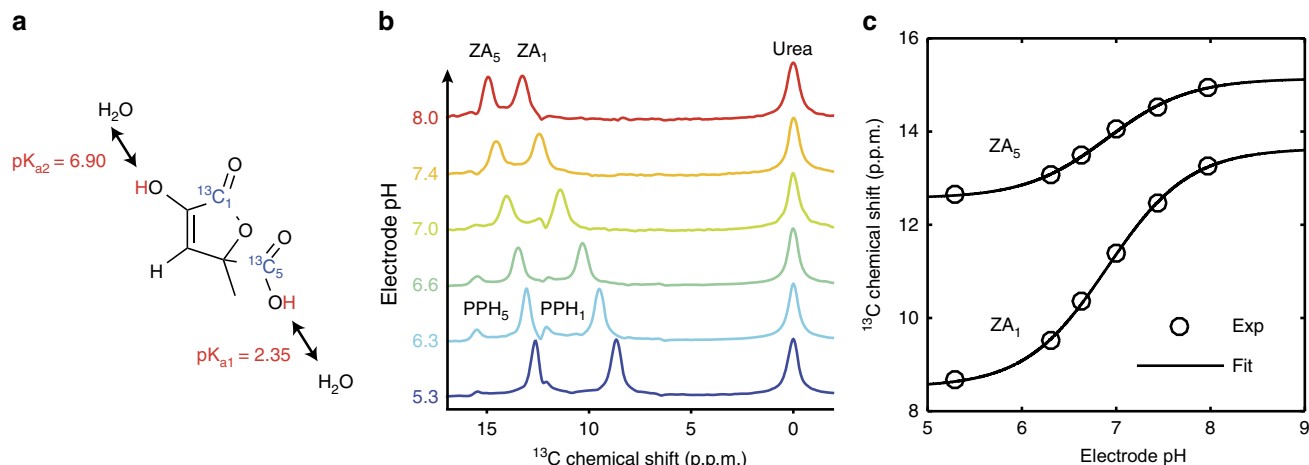

**Figure 1 | Mechanism and calibration of the pH dependency of ZA at 7 T. (a)** ZA bears two exchangeable protons, one in a carboxy group with an acid dissociation constant $pK_{a1} = 2.35$ and a second one in an enolic hydroxy group with $pK_{a2} = 6.90$. **(b)** Spectra of hyperpolarized ZA and urea in buffer phantoms at different pH values. The pH sensitivity in the physiological range is due to $pK_{a2}$ and results in a change of the chemical shifts of the two $^{13}C$-labelled carbon positions with respect to the pH insensitive $^{13}C$ urea peak as a function of pH. Additional peaks of a decay product of ZA, PP hydrate (PPH), can be identified. **(c)** The fit of a scaled logistic function to the data from **b** results in a calibration curve with a direct dependence between chemical shift and pH.

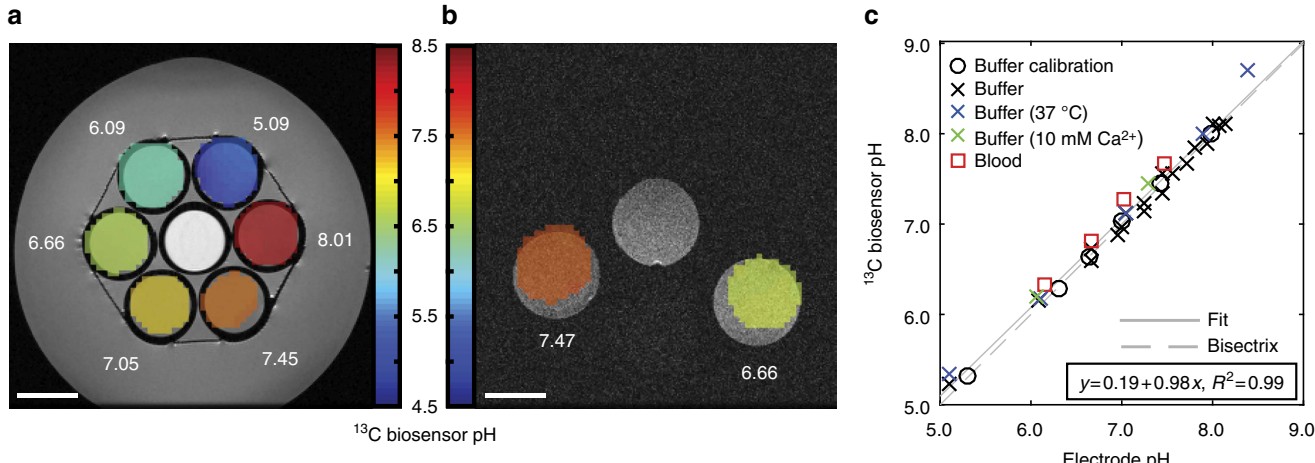

**Figure 2 | Validation of ZA as a pH sensor in hyperpolarized *in vitro* measurements in buffer and blood at 7 T. (a,b)** Hyperpolarized $^{13}C$ biosensor pH maps (coloured) in an axial slice overlayed on proton images (greyscale). The electrode pH values are shown in white. ZA and urea were simultaneously hyperpolarized and injected into **(a)** six buffer phantoms and **(b)** two blood phantoms prepared at different pH values. In both **a,b**, a calibration phantom (see Methods section) was present. $^{13}C$ biosensor pH map windows are based on a signal-level threshold. **(c)** $^{13}C$ biosensor pH and electrode pH correlate well for experiments in different buffers (black circles and crosses), at elevated temperature (37 °C, blue crosses), at a 10 mM $Ca^{2+}$ concentration (green crosses) and in blood (red squares). We determined a root mean square error of the fitting residuals (RMSE) of 0.10 pH units from the set of all *in vitro* experiments shown in Fig. 2c. Scale bars, 1 cm.

Furthermore, we showed that variations in ionic strength (Supplementary Fig. 12) and protein concentration (Supplementary Fig. 13) are not substantially influencing ZA's pH determination accuracy.

The acquisition of spatially localized pH maps *in vivo* requires a sufficiently long signal lifetime. Longitudinal relaxation times were determined *in vivo* by performing slice selective time-resolved spectroscopy, resulting in an apparent $T_1$ of $17 \pm 2$ s (mean ± s.d.; $n = 6$, three independent slices per two animals) for $ZA_1$ and $16 \pm 1$ s (mean ± s.d.; $n = 6$) for $ZA_5$ at 7 T (Supplementary Fig. 14). Notably, the apparent $T_1$ of ZA is slightly longer than the apparent $T_1$ of urea ($13 \pm 2$ s, mean ± s.d., $n = 6$, $B_0 = 7$ T), placing ZA on the same level with a well-established hyperpolarized compound.

***In vivo* pH measurement in rat bladder.** First, we tested whether a pH measurement using ZA gives reliable results *in vivo* where an independent validation of the pH is possible. To this end, we injected simultaneously hyperpolarized ZA and urea via a catheter directly into the filled rat bladder. The source of the hyperpolarized signal of both ZA and urea (Fig. 3a,b) clearly stemmed from the bladder. From the localized spectroscopic imaging data (Fig. 3d), a pH map (Fig. 3c) was calculated, resulting in a $^{13}C$ biosensor pH of $6.48 \pm 0.02$ ($n = 9$ pixels, mean ± s.d.), in good agreement with the pH of $6.55 \pm 0.01$ determined via pH electrode in the urine, sampled from the bladder directly after the MR measurement. All $^{13}C$ images in animals were acquired with a nominal spatial resolution of $3.75 \times 3.75 \times 5$ mm$^3$.

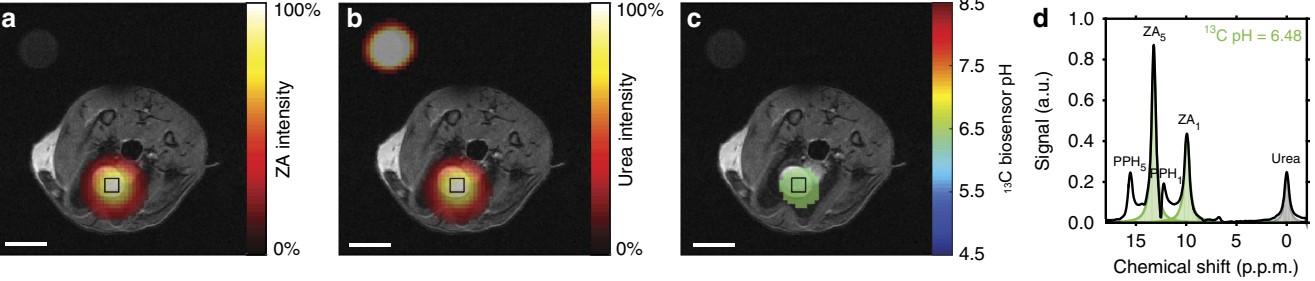

**Figure 3 | Hyperpolarized ZA *in vivo* pH measurement in the rat bladder at 7 T.** (**a–c**) Hyperpolarized $^{13}$C measurements (coloured) in an axial slice overlayed on anatomical proton images (greyscale). A calibration phantom containing $^{13}$C urea is visible in the upper left corner of the image. The signal stemming from the two simultaneously hyperpolarized and injected substances ZA (**a**) and urea (**b**) is well localized to the bladder. (**c**) The pH map calculated from changes in chemical shift of the $^{13}$C pH biosensor shows an acidic pH in the bladder of $6.48 \pm 0.02$ ($n = 9$ pixels, mean $\pm$ s.d.). (**d**) The spectrum from a voxel at nominal resolution (black box) in the bladder shows ZA (fitted, green), urea (fitted, grey) and a noticeable amount of a decay product of ZA, PPH (not fitted, white). Scale bars, 1 cm.

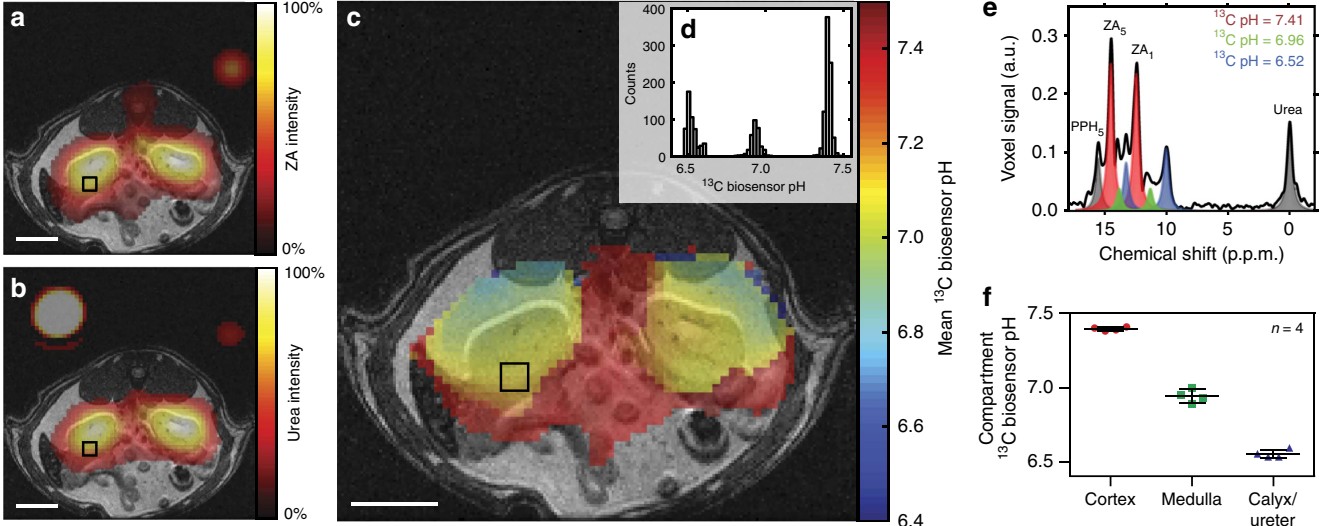

**Figure 4 | Hyperpolarized ZA *in vivo* pH measurements show three pH compartments in rat kidneys at 7 T.** Representative kidney data from a hyperpolarized $^{13}$C measurement (coloured) in an axial slice overlayed on anatomical proton images (greyscale). A calibration phantom containing $^{13}$C urea and the catheter used for injection are visible. The two simultaneously hyperpolarized and injected substances (**a**) ZA and (**b**) urea show high signal intensities in both kidneys of a healthy rat. (**c**) The mean pH map shows lower pH values in the kidneys compared to the surrounding tissue. (**e**) A voxel can contain up to three pairs of ZA peaks (red, green and blue) and a noticeable amount of PPH. The pH values group into three clusters (**d**, shown for one representative animal), consistently demonstrated in four animals (**f**, individual data points and mean $\pm$ s.d.). For all $^{13}$C images, intensity windows are based on sufficiently high signal levels for either (intensity images) or both (pH images) ZA and urea. Scale bars, 1 cm.

**In vivo pH measurement in rat kidneys.** Next, a tail vein injection of hyperpolarized ZA and urea into four healthy rats was performed, subsequently imaging a slice containing the kidneys. Starting the imaging sequence 10 s after completion of the injection, the strongest signal from hyperpolarized ZA and urea originated from the kidneys with a noticeable amount detected in the blood pool (Fig. 4a,b). We discovered up to three pairs of ZA peaks within the same voxel (Fig. 4e) forming three pH clusters with a pH of $7.40 \pm 0.01$, $6.94 \pm 0.05$ and $6.55 \pm 0.03$ (Fig. 4d,f, $n = 4$ rats, mean $\pm$ s.d., representative fitting residuals shown in Supplementary Fig. 15). Based on their consistent pH values with data from literature[22], these pH clusters were tentatively attributed to the cortex, medulla and calyx/ureter, respectively, reflecting the transition effects of the blood pH before, during and after processing, filtering and renal metabolism. Weighing the pH from the three compartments by their respective amplitudes, a mean pH map was calculated (Fig. 4c). The extravasation of ZA and its accumulation in renal tissue in both cortex and medulla was confirmed by matrix-assisted laser desorption/ionization mass spectrometry imaging (MALDI-MSI, Supplementary Fig. 16).

**In vivo pH measurement in rat tumours.** Finally, we imaged the pH within MAT B III adenocarcinoma in five rats. Starting the measurement 10 s after the end of the intravenous injection of the hyperpolarized compounds, the strong ZA and urea signals showed a well-perfused subcutaneous tumour (Fig. 5a,b, additional proton images shown in Supplementary Fig. 17). We detected an acidic tumour compartment of pH $6.94 \pm 0.12$ ($n = 5$ rats, mean $\pm$ s.d.) tentatively assigned to the extravascular (interstitial/extracellular) compartment of the tumour. The signal stemming from the other tumour compartment of pH $7.40 \pm 0.05$ ($n = 5$ rats, mean $\pm$ s.d.) was tentatively assigned to the vasculature of the tumour. Furthermore, we observed a pH of $7.39 \pm 0.05$ ($n = 5$ rats, mean $\pm$ s.d.) in blood near the vena cava (Fig. 5c–e). These measured tumour pH values were compared to three other independent interstitial/extracellular pH measurement methods in MAT B III tumour bearing rats

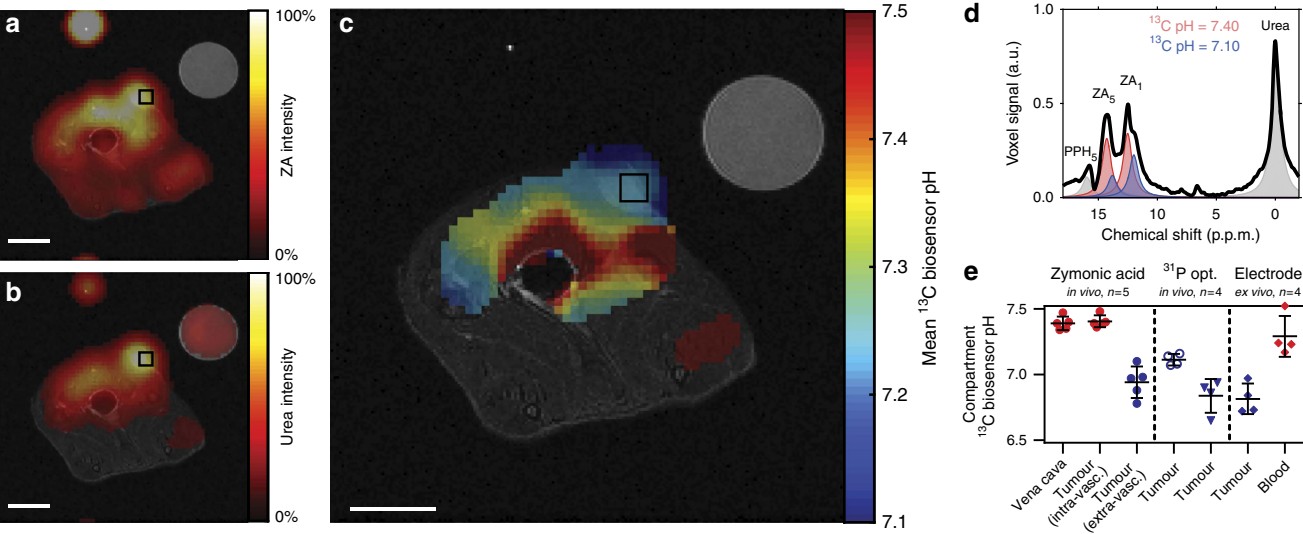

**Figure 5 | Hyperpolarized ZA in vivo pH measurements show an acidic tumour pH at 7 T.** Representative tumour data from a hyperpolarized [13]C measurement (coloured) in an axial slice overlaid on anatomical proton images (greyscale). A calibration phantom containing [13]C urea and the catheter used for injection are visible. Both (**a**) ZA and (**b**) urea accumulate in the MAT B III tumour. (**c**) The mean pH map shows a lower pH value in the tumour compared to the surrounding tissue. For all [13]C images, intensity windows are based on sufficiently high signal levels for either (intensity images) or both (pH images) ZA and urea. Spectra from tumour voxels (**d**, shown for one representative animal) are best fitted with two pairs of ZA peaks (red, blue) and consistently show different pH values in the tumour in comparison to the vena cava, demonstrated in five animals (**e**, individual data points and mean ± s.d.) and compared with three other interstitial/extracellular pH measurements in four animals, namely [31]P MRS in vivo using 3-APP, a needle-type optical sensor in vivo and an ex vivo tissue electrode pH measurement. Scale bars, 1 cm.

($n = 4$ rats, mean ± s.d.): First, the extracellular [31]P pH probe 3-aminopropylphosphonate (3-APP)[23,24] showed an extracellular pH of $7.11 \pm 0.04$ in vivo; second, the pH determined using a needle-type optical sensor was $6.84 \pm 0.13$ in vivo; third, ex vivo measurement of tumour tissue using a standard pH microelectrode gave a pH of $6.82 \pm 0.12$ (Fig. 5e). These three pH values agree well with the value determined for the extravascular (extracellular/interstitial) pH compartment using ZA (no significant difference, $P > 0.05$, Mann–Whitney–Wilcoxon test) suggesting that ZA measures and distinguishes extravascular (interstitial/extracellular) pH and intravascular pH.

## Discussion

Despite the relevance of extracellular pH for the characterization of human pathologies, there is currently no method to non-invasively map the spatial distribution of pH by clinical imaging methods, creating a pressing need for the development of pH imaging probes with potential for clinical translation. Here we present a novel pH imaging approach using hyperpolarized ZA as an MRI pH sensor. Our results showed that this pH imaging method is independent of concentration, temperature, ionic strength and protein concentration. Importantly, ZA's pH sensitivity arises from protonation and deprotonation in proximity to its [13]C-labelled sites, which is fast on the NMR timescale ($10^3$–$10^4$ Hz under physiological conditions[25]) and allows an assessment of pH whenever the agent reaches its target. We demonstrated that in vivo pH maps within rat kidneys and adenocarcinomas can reliably be determined, and in both cases two to three pH compartments within one imaging voxel can be separated, which is a unique feature of ZA. Important criteria for the evaluation of a novel pH probe are the achievable spatial and temporal resolution, its sensitivity, accuracy to determine pH, toxicity, its potential to disturb the acid–base balance during the measurement, and knowledge about its presence in the vascular, extravascular or intracellular compartments or a combination of the aforementioned.

In comparison to hyperpolarized [13]C-labelled bicarbonate[26], which has been proposed as a probe for clinical pH imaging and which currently is the most prominent among the hyperpolarized pH probes, ZA-based pH imaging offers several advantages: First, ZA has a relatively long signal lifetime with an apparent $T_1$ of $\approx 17$ s at 7 T in vivo, whereas bicarbonate has a shorter apparent $T_1$, which is $\approx 10$ s in vivo at both 3 T and 9.4 T (refs 26,27); second, ZA is highly soluble and does not need enzymatic conversion to become an active pH probe, whereas bicarbonate exhibits a relatively small equilibrium $CO_2$ signal ($\approx 6\%$ of $HCO_3^-$ at pH 7.4) and limited solubility at physiological pH; third, ZA is localized in the extracellular space and slowly excreted through the kidneys and decomposes into endogenous substrates, whereas $CO_2$ through respiration and diffusion across cell membranes is not clearly restricted to the extracellular space; fourth, ZA relies on the determination of pH through the measurement of chemical shift displacements, whereas for bicarbonate pH mapping, there is a need for ratiometric analysis of signal amplitudes which is difficult at a low signal-to-noise-ratio (SNR); fifth, pH measurements using ZA does not involve any enzymes in contrast to bicarbonate pH imaging, where the enzyme concentration (that is, carbonic anhydrase) influences the speed at which pH can be measured.

The pH determination accuracy of ZA was found to be high enough to detect relevant pathological pH changes in vivo. The pH dispersion of the [13]C chemical shift of $ZA_1$ in the physiological pH range is $\approx 3.0$ p.p.m. per pH. This corresponds to a frequency shift of 96 Hz per pH at a clinical field strength of 3 T. Assuming that peaks can be distinguished at 20% of their linewidth apart, which we measured to be on the order of 30–50 Hz in vivo, this results in a pH resolution of $\approx 0.1$ pH units. Similar accuracies have been reported for other chemical shift-based hyperpolarized probes like [13C,15N]ACES, although this probe has so far only been proven to work in vitro[12]. A larger pH dispersion of the [13]C chemical shifts of a probe would theoretically allow for a higher accuracy of the pH measurement but comes with the drawback that RF excitation pulses then need

to cover a larger bandwidth, which is challenging for the RF pulse design. At the same time, chemical shift displacement artifacts impede accurate slice selective excitation of all pH-sensitive resonances if resonances are spread out further. In addition, the chemical shift imaging (CSI) readout would need to cover a larger frequency bandwidth resulting in increased noise from the receiver. These limitations could pose a challenge for the recently introduced hyperpolarized [15]N-labelled probes based on pyridine and histidine derivatives that have been reported to show pH-dependent chemical shifts of more than 80 p.p.m.[11,28]. Also, the applicability of these probes still has to be proven under in vivo conditions. Although our fitting residuals shown in Supplementary Fig. 15 indicate that fitting ZA in multiple tissue compartments in kidney and tumour is reasonable, complete separation and fitting of extravascular and intravascular pH compartments remains challenging for the case of small pH differences between individual compartments.

A limitation for most hyperpolarized [13]C imaging methods is the low spatial resolution compared to other imaging methods. Spectral bleeding artefacts, where high-intensity spectral peaks are reproduced in neighbouring voxels, can lead to a further reduction in the resolution. Chemical exchange saturation transfer imaging methods have a better spatial resolution in the μm–mm range with good specificity but poor sensitivity[7]. The FIDCSI sequence used in this work (see Methods section) has a slice thickness of 5 mm and a nominal in-plane resolution of 3.75 mm. Zero filling by a factor of four results in an apparent in-plane resolution of 0.94 mm. Taking into account the shape of the point spread function of the circular sampling scheme, the in-plane resolution is affected by a factor of $\approx 1.40$ resulting in a real in-plane resolution of 5.25 mm and an apparent in-plane resolution of 1.31 mm. The spatial resolution of hyperpolarized [13]C pH imaging could be further improved by implementing alternative acquisition schemes such as echo planar spectroscopic imaging or using cryogenically cooled receive coils such that sub-millimetre resolution for pH imaging seems possible. Echo planar spectroscopic imaging increases the gradient hardware and magnetic field homogeneity requirements but reduces the time needed to acquire an image, thereby improving SNR and real resolution[21].

We have established non-toxicity of our agent in a toxico-pathological study and with dose escalation experiments at doses up to five times of the injected dose for the in vivo pH measurement. This might facilitate clinical translation compared to other pH imaging agents, where toxic effects can arise from $Gd^{3+}$-chelates in patients with renal insufficiency[29] or from preparations of hyperpolarized caesium bicarbonate, where toxic caesium[30] is used to increase the bicarbonate concentration.

The injected substances ZA and urea could potentially act as buffering agents and influence the pH value they are measuring. With a $pK_a \approx 0.18$ of its conjugated acid in water, urea cannot act as a buffer in the physiological pH range and thus does not influence the measured pH. For all pH probes for which the pH determination is based on a protonation/deprotonation mechanism within the pH range of interest, the pH measurement is effectively based on the buffering functionality of the pH probe. ZA with a $pK_a$ of 6.90 is injected at a concentration of 50 mM and is diluted to a concentration of about 5 mM if evenly distributed in the blood. The actual bicarbonate concentration in the blood at full oxygen saturation and 37 °C is 21–26 mM, with the $CO_2/HCO_3^-$ equilibrium acting as the main natural buffer component in human. When, as in the current study, the injection of ZA results in low mM concentrations of ZA in the blood, one should not expect a significant alteration in pH even if much larger relative doses[31] are injected in vivo.

Evidence suggests that ZA extravasates and predominantly stays in the extracellular/interstitial space, thus measuring extracellular pH. Extravasation of ZA to renal tissue is confirmed by MALDI-MSI in kidney slices (Supplementary Fig. 16). Comparison of the pH measured using ZA in MAT B III tumour bearing rats with the pH values determined by three other interstitial/extracellular pH measurements ([31]P MRS, needle-type optical sensor, electrode) showed good agreement between these methods, suggesting that ZA can measure and distinguish both extravascular (interstitial/extracellular) and intravascular pH (Fig. 5e). Furthermore, the fact that ZA, instead of using a ratiometric method, uses differences in chemical shifts to determine the pH, allows for the detection of several pH compartments within the same voxel (Fig. 4e). If an intracellular pH compartment with a pH predominantly around 7.4 would provide a significant contribution to the overall signal, an additional pair of ZA peaks should be observed. Supplementary Fig. 1 shows a cell experiment with just one pair of ZA peaks belonging to a compartment with a [13]C biosensor pH that correlates well with the pH measured using a conventional pH electrode measuring the extracellular pH. Since no second pair of ZA peaks is visible in this experiment, no significant amount of ZA has entered the cell suggesting that ZA stays mainly extracellular.

ZA can be used for pH imaging either with simultaneously hyperpolarized urea as a pH-independent chemical shift reference as demonstrated in this work or without urea to decrease the number of substances required to be injected in vivo. When ZA is used as a pH imaging sensor without urea, the difference in chemical shift as a function of pH between the two [13]C positions within ZA can be used to back calculate the pH (Supplementary Fig. 5). Within the range of pH 6–8, both methods give nearly identical results, but the dynamic range of potential pH values to be measured is larger when the urea peak is included in the pH evaluation, since the pH determination precision is higher at the limit of the sensor's sensitivity when using urea as a reference (RMSE $\approx 0.09$ pH units compared to RMSE $\approx 0.019$ pH units between the [13]C biosensor pH measurement and the electrode pH measurement).

As for other currently known [13]C-labelled probes, a major limitation of using hyperpolarized ZA for in vivo applications is its polarization lifetime. This could be prolonged by performing the pH measurements at a reduced clinical field strength and by deuteration. For hyperpolarized [13]C experiments, higher magnetic fields result in a larger chemical shift separation between the peaks of interest, allowing for easier discrimination. However, higher magnetic fields lead to shorter $T_1$ relaxation times, reducing the time available for hyperpolarized measurements, and in contrast to conventional experiments, higher magnetic fields do not result in larger signals. Thus, it would be best to perform the measurements at an intermediate and well-shimmed magnetic field, for example, at clinically widely available 3 T scanners, to balance the chemical shift separation and $T_1$ relaxation times. In addition, dipolar effects between the four protons and the two [13]C-labelled carbons reduce the $T_1$ relaxation time. Replacing both the single proton and the three protons in the methyl group of ZA with deuterium would lengthen the $T_1$s of $ZA_1$ and $ZA_5$. Sensitivity could be improved further by increasing the polarization level.

ZA represents a pH imaging candidate for clinical translation. The demonstrated robustness of using ZA for extracellular pH imaging, its non-toxicity and the diagnostic value of imaging pH in a broad pathological context, such as in ischaemia, infection, inflammation and cancer, render this new pH imaging technique valuable. Changes in metabolism and extracellular pH have been shown to be an indicator of early treatment response in tumours preceding morphologic transformations[32], suggesting that ZA could be used to detect pH changes as an early biomarker of successful therapy.

## Methods

**Chemicals.** [13]C- and [2]H-enriched chemicals were either purchased from Sigma-Aldrich (USA) or Euriso-Top (France). Unless stated differently, all other chemicals were purchased from Sigma-Aldrich (USA).

**Synthesis of selectively [13]C-labelled ZA.** Equal amounts (v/v) of [1-[13]C]pyruvic acid and 37% hydrochloric acid were incubated for 20–30 days[18]. Optionally, 1 eq. [1-[13]C]pyruvic acid was incubated with 0.5 eq. aqueous zinc acetate solution for 1 h yielding insoluble [1,5-[13]C$_2$]zinc PP, which was dissolved with 37% hydrochloric acid. After the removal of volatiles using a rotatory evaporator, the residual yellow viscous solution was dissolved in a small amount of water containing 0.1% trifluoroacetic acid. The reaction product was purified by a reversed phase high pressure liquid chromatography using a Waters XBridge Prep C18 column (linear gradient of 2–20% acetonitrile, buffer A: 0.1% TFA in water, buffer B: 0.1% TFA in acetonitrile) and freeze-dried *in vacuo* (alpha 2–4 LD plus, Christ, UK). The experimental yield of ZA was ∼ 40%. ZA can be stored lyophilized and is stable in dimethylsulphoxide.

**Cell experiments.** MCF-7 cells used in this study were originally provided by Guy Leclerq at the Institut Jules Bordet, Centre des Tumeurs de l'Université Libre de Bruxelles, Belgium, in whose lab they have since been cultured and tested as free of mycoplasma. The identity of the cells was confirmed by the Leibniz-Institut Deutsche Sammlung von Mikoorganismen und Zellkulturen (DSMZ), Germany in 2015. MCF-7 tumour cells were maintained as a monolayer culture in Dulbecco's modified Eagle's medium supplemented with 5% foetal calf serum. For spheroid preparation, cells were gently stirred in spinner flasks for up to 6 days in a CO$_2$ incubator at 37 °C and transferred to Dulbecco's modified Eagle's medium with 5% foetal calf serum containing NaHCO$_3$ (0.37 g l$^{-1}$) and 20 mM 2-[4-(2-hydroxyethyl)piperazin-1-yl]ethanesulfonic acid at pH 7.4. After gradually inducing cell membrane permeabilization, and thus cell death and medium acidification by incubation with 0.015% Triton X-100, 0.2 ml of a 10 mM hyperpolarized [1-[13]C]pyruvic acid solution containing traces of hyperpolarized ZA were added to 0.8 ml of a cell spheroid suspension containing $40 \times 10^6$ MCF-7 cells in assay medium. [13]C NMR spectra were acquired at 14.1 T (Avance III, Bruker BioSpin, Germany). The [13]C biosensor pH was calculated from the difference in chemical shift of the two peaks assigned to the [13]C-labelled positions (ZA$_5$ and ZA$_1$) in ZA using the model shown in Fig. 1c and compared to the pH determined after the NMR experiment with a standard pH electrode (pH metre: ProLab 4000, SI Analytics, pH electrode: N6000A, temperature sensor Pt 1000: W5790NN). The fraction of dead cells[13] was determined by centrifugation of the spheroid suspension at 150$g$ before washing the pellet twice with phosphate buffered saline (PBS) and incubating an aliquot of spheroids at room temperature for 30 min with a solution containing 2 μM membrane-permeant ethidium bromide. After another centrifugation, spheroids were fixed in 3.7% formaldehyde PBS solution, analysed by fluorescence microscopy (Axiovert 200, Zeiss, Germany) using fluorescein isothiocyanate and rhodamine filters to discriminate between general green autofluorescence and red staining of dead cells. MTT assays to assess cellular metabolic activity were performed by incubating $5 \times 10^3$ HeLa cells, each suspended in 100 μl cell culture medium at pH 8.0, for 24 h with ZA at concentrations between 0.4 and 12.5 mM.

**Characterization of ZA.** The mass of the synthesized substance was determined by an HR-MS-spectrum using a Thermo Finnigan LTQ-FT. A MS/MS-spectrum was recorded by CID-fragmentation on a Thermo Finnigan LCQ-Fleet (Supplementary Fig. 2). [1]H and [13]C chemical shifts and coupling networks were extracted from NMR spectra of unlabelled and [13]C fully labelled ZA and its decay products in H$_2$O at 14.1 T (Supplementary Fig. 1). ZA concentration-dependent [13]C NMR acid-base titrations were performed with ZA at 37 °C in 1 M phosphate buffer at 14.1 T, adjusting the pH in random order using 1 M HCl and 1 M NaOH. Electrode pH was reported as the mean of two measurements determined directly before and after the NMR experiment with a standard pH electrode. Regular acid-base titrations of 25 mM ZA in 1 M KCl in H$_2$O were performed using 1 M KOH at different temperatures. Each full titration was performed within $22 \pm 1$ min and a model titration curve for a diprotic acid was fitted to each titration experiment separately to extract the relevant acid dissociation constant pK$_{a2}$ as a function of temperature. The temperature dependence of the acid dissociation constant $\Delta$pK$_{a2}/\Delta$T was then determined from a linear fit to the extracted data points. Hyperpolarized [13]C NMR acid-base titrations were performed with a solution of 25 mM ZA in presence of 10 mM Ca$^{2+}$ at 1 T and 27 °C. The detected pH-dependent hyperpolarized [13]C chemical shifts were compared to regular [13]C NMR acid-base titrations at 37 °C in 1 M phosphate buffer at 14 T. Stability measurements were performed by acquiring [1]H spectra at 1 T and 27 °C over 20 h, each spectrum averaged over 60 scans with a repetition time of 10 s.

**Hyperpolarization with dissolution DNP.** A solution of 50 μl of dry dimethyl-sulphoxide with 4 M ZA, 15 mM of free radical (OX063, Oxford Instruments) and 5 mM of gadolinium chelate (Dotarem, Guerbet) was added to a standard DNP sample cup and frozen in liquid nitrogen. An aliquot of 30 μl 10 M [13]C urea containing 30 mM of free radical and 1.5 mM of gadolinium chelate were added on

top of the frozen ZA layer and frozen in liquid nitrogen as well. The sample cup was placed in a DNP polarizer (Hypersense, Oxford Instruments, UK) and polarization was carried out using a microwave source at 94.155 GHz for 1.5 h. The sample was dissolved in a pressurized (10 bar) and heated (180 °C) solution of 4 ml D$_2$O containing 0.1 g l$^{-1}$ sodium diaminoethanetetraacetic acid (Na$_2$EDTA), which was neutralized with NaOH to a final pH of ∼ 7.4. The final concentrations of ZA and [13]C urea were ∼ 50 mM and 75 mM. The final temperature of the solution was 37 °C. $T_1$ measurements of natural abundance ZA in 80 mM Tris buffer in H$_2$O were performed on a clinical Biograph mMR MR-PET (Siemens, Germany, $B_0 = 3$ T) using a pulse length of 0.2 ms, a flip angle of 15°, a spectral bandwidth of 2,500 Hz and a repetition time of 5 s. Carbon centre frequency and flip angle were determined using an 8 M [13]C urea phantom with 5 mM Dotarem (Guerbet, France) and 0.1% sodium azide. $T_1$ decay curves were flip angle corrected and fitted by a three-parameter mono-exponential decay curve. The solution polarization level was determined using a Bruker Minispec mq40 NMR analyser (Bruker, Germany, $B_0 = 1$ T). First, a hyperpolarized $T_1$ decay curve was acquired, flip angle corrected, and the signal intensity was extrapolated to the time of dissolution. The thermal signal was measured with a Carr–Purcell–Meiboom–Gill sequence (echo time: 2 ms, acquisition time: 0.25 ms, repetition time: 300 s, number of echoes: 600, number of scans: 10).

**Buffer and blood phantom preparation for imaging experiments.** Buffer phantoms were either prepared using a 200 mM disodium phosphate/sodium phosphate buffer, an 80 mM Tris buffer or a 100 mM citric acid/200 mM disodium phosphate buffer. They were placed into a water bath for improved shimming and reduced susceptibility artefacts. For experiments at 37 °C, an air heating unit with feedback control (SA Instruments, USA) was used to stabilize the phantoms' temperature. An aliquot of 0.2 ml of hyperpolarized ZA and urea solution were added to each 2.0 ml buffer phantom using a multistep pipette. Phantoms were closed and inverted three times for thorough mixing before placing them into the MR scanner. Blood phantoms were prepared with human blood withdrawn into collection tubes containing EDTA as anticoagulant (S-Monovette, EDTA KE, Sarstedt, Germany) to maintain blood in its fluid state and the blood pH was adjusted using 1 M HCl. Phantoms were placed inside the MR scanner with catheters inserted for injection of 0.2 ml of hyperpolarized ZA and urea solution. Direct and rapid injection into each 1.8 ml blood phantom prevented fast $T_1$ relaxation of ZA in blood at Earth's magnetic field[33]. A thorough mixing of the solution was achieved by shaking the entire phantom set-up. For both buffer and blood phantom experiments, an additional [13]C urea phantom doped with a gadolinium chelate (Dotarem, Guerbet, France) was placed inside the field of view for [13]C flip angle and frequency calibration. The pH of each phantom was measured by a standard pH electrode after the MR experiment.

**Spectroscopy and imaging procedure.** All phantom and [13]C *in vivo* imaging experiments were performed on a 7 T small animal MR scanner (GE, Agilent), which was migrated to a Bruker BioSpec console before [31]P experiments were performed. A 72 mm dual tuned [1]H/[13]C birdcage coil was used for signal transmission in combination with a two channel flexible coil [13]C receive array (Rapid Biomedical, Germany). Proton images were recorded with a fast spin echo sequence, slice thickness 1 mm, field of view 6 cm, image matrix $256 \times 256$, repetition time 2.6 s, effective echo time 20 ms and number of averages 2. *In vivo* $T_1$ relaxation was measured using slice selective excitation with 10 mm slice thickness, 6 cm field of view, 10° flip angle, 3 s repetition time, 64 excitations and total scan time 192 s. The acquired free induction decays were zero filled, line-broadened by an exponential filter of 20 Hz and Fourier transformed using Mnova (Mestrelab Research, Spain). A three-parameter mono-exponential decay curve was fitted in Matlab (MathWorks, USA). Imaging experiments were performed using a FIDCSI sequence and reconstruction in Matlab (MathWorks, USA) described elsewhere[21], with k-space being sampled in centric order from the centre on outwards. Imaging parameters were: 141 T m$^{-1}$ s$^{-1}$ maximum slew rate, 38 mT m$^{-1}$ maximum gradient amplitude, 5 kHz spectral bandwidth, 512 points sampled, 9.8 Hz spectral resolution, $16 \times 16$ nominal matrix size, 5 mm slice thickness, 6 cm field of view, 4° flip angle, 118 ms repetition time, 208 excitations and total scan time 25 s. A 10 Hz Gaussian filter was applied along the readout dimension and the data were zero-filled by a factor of four in the two spatial as well as in the spectral dimension before fast Fourier transforms were applied along all three dimensions. For all [13]C images, intensity windows are based on sufficiently high signal levels for either (intensity images) or both (pH images) ZA and urea.

**Animal handling and tumour implantation.** Animal experiments were approved by the local governmental committee for animal protection and welfare (Tierschutzbehörde, Regierung von Oberbayern) and performed in accordance with the institutional guidelines of the Technische Universität München for the care and use of animals. Animals were anaesthetized with 3–5% isoflurane inhalation gas. Catheters were placed before the animals were put into the MR scanner and tail vein injections were performed at a dose of 5 ml kg$^{-1}$ and a rate of 0.17 ml s$^{-1}$. MR exams were started 10 s after the end of the injection. To maintain constant body temperature, animals were placed on an electric heating pad during the dose escalation study, and on a water heated pad during MR exams. In the

dose escalation study, heart rate and breathing rate (ECG Trigger Unit, Rapid Biomedical, Germany), as well as blood oxygenation (PalmSAT 2500A VET Pulsoximeter, Nonin, USA) were monitored. The dose escalation study was carried out with three male rats ($\sim$12 weeks, Lewis, Charles River, average weight $319 \pm 1$ g), the in vivo $T_1$ experiments with two male rats ($\sim$8 weeks, Lewis, Charles River, average weight $217 \pm 1$ g), the bladder experiment was carried out with a female rat ($\sim$8 weeks, Lewis, Charles River, weight 230 g), the kidney experiment with four male rats ($\sim$14 weeks, Buffalo, Charles River, average weight $375 \pm 38$ g), the tumour experiment with five female rats ($\sim$6 weeks, Lewis, Charles River, weight $132 \pm 10$ g), the MALDI experiment was carried out with a male rat ($\sim$8 weeks, Wistar, Charles River, weight 300 g), the phosphor and optical pH measurements with four female rats ($\sim$9 weeks, Fischer 344, Charles River, weight $155 \pm 2$ g) and the electrode pH measurements with four female rats ($\sim$8 weeks, Fischer 344, Charles River, weight $153 \pm 2$ g). For the bladder experiment, saline (0.90% w/v of NaCl) was injected subcutaneously 30 min before the start of the experiment to guarantee accumulation of urine in the bladder. A catheter for injection of the hyperpolarized substances and for sampling the urine was inserted into the bladder before placing the animal in the MR scanner. Urine pH was measured using a standard pH electrode. Tumour imaging experiments were performed on day 6 after injecting 200 µl PBS containing $1 \times 10^6$ 13762 MAT B III (ATCC, USA) tumour cells subcutaneously into the right flank. Cells were tested for mycoplasma with negative result. Animal MRI experiments were not randomized and not blinded.

**Toxicopathological study.** To examine potential acute and/or subacute toxicological effects of ZA, we performed a toxicopathological study in Fischer 344 rats obtained from Charles River with 6 weeks of age (average weight $150 \pm 8$ g). In total, 14 animals (7 female/7 male) were used. Ten of them received a tail vein injection of 5 ml kg$^{-1}$ with a concentration of 250 mM ZA (five times the dosage used for the imaging experiments), four animals served as controls with a tail vein injection of saline (0.09% w/v of NaCl). Six animals were killed 24 h after injection (acute toxicity), eight animals after 4 weeks (subacute toxicity). The animals were monitored daily, blood collection for haematology and clinical chemistry was performed before and 24 h (acute), 7, 21 and 30 days (subacute) after injection of ZA. The following parameters were measured: haematocrit, haemoglobin, erythrocyte and total leucocyte counts and differentials, urea, creatinine, total protein, aspartate aminotransferase, alkaline phosphatase, creatine kinase, gamma glutamyl transferase, glutamate dehydrogenase, fructosamines, calcium, potassium, magnesium, sodium, inorganic phosphate and bile salts. After killing, a necropsy was performed, organs (liver, kidneys, spleen, pancreas, intestines, lymph nodes, injection sites, genital tract, lung, heart, skin, skeletal muscle and brain) were fixed in 10% neutral buffered formalin overnight, dehydrated according to standard protocols, embedded in paraffin, cut in 2 µm thick sections and routinely stained with haematoxylin-eosin (H&E). H&E stainings of all organs were analysed blinded by an experienced animal pathologist (K.S.) and findings were reported according to the INHAND criteria of the Society of Toxicologic Pathology with respect to the most recent recommendations. Sample size estimates were not performed since both mean and s.d. had to be determined in the experiment for each group.

**Back-calculation of pH values and maps.** Data analysis was carried out using Matlab (MathWorks, USA) on a voxel-by-voxel basis. Magnitude signals of the two coils were combined assuming equal coil sensitivities[34]. Peaks were detected and fitted to a sum of Lorentzian functions based on their chemical shift with respect to urea (165.5 p.p.m.), which was used as a chemical shift reference set to 0 p.p.m.. During the fitting procedure, peak positions and amplitudes were varied while the peaks' linewidth was determined by the urea peak linewidth ($50 \pm 13$ Hz (mean $\pm$ s.d. from all imaging experiments shown)). PP hydrate peaks were identified according to their expected chemical shifts. For each pair of ZA peaks, a pH value was calculated by a nonlinear least squares algorithm minimizing the residual between the experimental and modelled chemical shift differences of $ZA_5$ and $ZA_1$ simultaneously with respect to urea. In case of fast chemical exchange, both chemical shifts $ZA_5$ and $ZA_1$ can be described as a function of pH by a scaled logistic function $ZA_i$ (pH) $= ZA_{i,min} + \delta_i/(1 + 10^{(pK_a - pH)})$ with the same acid dissociation constant $pK_a$ for both curves, resulting in five parameters derived simultaneously from the two calibration curves, $ZA_{5,min} = 12.57$ p.p.m., $ZA_{1,min} = 8.52$ p.p.m., $\delta_5 = 2.57$ p.p.m., $\delta_1 = 5.13$ p.p.m. and $pK_a = 6.90$. In case that several pairs of ZA peaks were found within the same voxel, a mean pH map was calculated weighted by the amplitudes of the respective pair of ZA peaks (as seen in the kidneys), in case where $ZA_1$ overlapped with $PPH_1$, so that a discrimination of the two was made impossible, only the shift difference of $ZA_5$ with respect to urea was used to calculate the mean pH map (as seen in the tumour). pH values were only calculated when required ZA peaks and urea were detected at an SNR of at least 10.

**MALDI mass spectrometry imaging.** Frozen tissue samples were cryosectioned into 12 µm thick slices and thaw-mounted to pre-cooled ($-20\,^\circ$C) conductive indium tin oxide-coated glass slides (Bruker Daltonics, Germany), pre-coated with a 1:1 mixture of poly-L-lysine and 0.1% Nonidet P-40. Tissue sections were coated with 10 mg ml$^{-1}$ 9-aminoacridine matrix in 70% methanol using a SunCollect sprayer

(Sunchrom, Germany). MALDI-MSI data were obtained using a Solarix 7 T FT-ICR MS (Bruker Daltonics). Mass spectra were acquired in negative mode for each position using 200 laser shots at a frequency of 500 Hz utilizing continuous accumulation of selected ions to increase signal intensities at m/z 157 with a window of 20 Da. After MALDI-MSI completion, the matrix was removed from slides by 70% ethanol, and tissue sections were stained using H&E.

**Phosphor pH reference measurements.** 3-APP was injected intra-peritoneal (3 ml of 64 mg ml$^{-1}$ in normal saline, pH adjusted to 7.4 using 10 M NaOH) 40 min before the acquisition of $^{31}$P spectra (3 s repetition time; 12 kHz spectral width; 4,096 acquisition points; 200 averages; 60° flip angle; 10 min acquisition time) using a 2 cm outer diameter double resonance $^1$H/$^{31}$P-loop coil (Bruker, Germany) placed on the subcutaneous tumours. 3-APP chemical shifts $\delta$ in $^{31}$P spectra were referenced to phosphocreatine and the pH was calculated from the Henderson–Hasselbalch equation pH $= pK_a - \log_{10}[(\delta - \delta_{min})/(\delta_{max} - \delta)]$ with $pK_a = 6.85$, $\delta_{max} = 27.01$ p.p.m., $\delta_{min} = 23.53$ p.p.m. (ref. 24).

**Optical and electrode pH reference measurements.** In vivo electrode pH measurements in tumours were performed using needle-type optical pH microsensors (NTH-HP5, PreSens, Germany) with a fibre optic metre (pH-1 micro, PreSens, Germany) and a manual micromanipulator (MM33, PreSens, Germany). Optical fibre electrodes were calibrated using a three-point calibration in blood with pH adjusted using HCl. Optical pH measurements were averaged over three different locations of the tumour performed during a stable signal reading for at least 3 min. For ex vivo electrode pH measurements in tumour and blood, blood was withdrawn from the tail vein into collection tubes containing EDTA while tumours were measured directly after extraction from the killed animal.

**Data availability.** The data supporting the findings of this study are available within the article and its Supplementary Information or from the corresponding author on reasonable request.

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

## Acknowledgements

We appreciated discussions with Wolfgang Eisenreich, Angela Otto, Raimund Marx and Simone Köcher. We thank Stephan Sieber for access to the preparative HPLC and mass spectrometers, Bernd Reif and his group for access to the lyophilizer and Wolfgang Heydenreuter and Franziska Mandl for their assistance in HPLC usage. Geoffrey Topping, Markus Durst and Ulrich Köllisch helped with regard to the imaging and spectroscopy sequences. Michael Michalik, Miriam Braeuer, Anna Bartels and Concetta Gringeri supported the animal MRI experiments. Olga Seelbach, Marion Mielke and Hilde Kalvelage assisted the histopathological measurements. We are grateful to Bernd Pichler for lending us his $^{31}$P/$^1$H surface coil. We thank GE Global Research Europe for granting access to the hyperpolarizer. We acknowledge support from EU Grant No. 294582 (MUMI), BMBF (FKZ 13EZ1114) and DFG (SFB 824).

## Author contributions

S.D. and F.S. identified ZA; S.D. and C.H. operated the polarizer and devised the recipe for hyperpolarization of ZA; S.D., C.H. and F.S. conducted the NMR and MRI experiments; C.H. and M.G. synthesized the compounds; M.G. performed the cytotoxicity assay; B.F. prepared the cells and animals; K.S., A.B. and A.W. performed pathological analyses; S.D. evaluated and analysed the data; S.D., C.H. and F.S. wrote the paper and all authors reviewed the manuscript; A.H., S.J.G., M.S. and F.S. designed the research; and F.S. devised the study.

## Additional information

**Competing interests:** S.D., M.G., S.J.G., and F.S. are named as inventors on a patent application concerning the use of ZA as a pH sensor molecule and declare competing financial interests. The remaining authors declare no competing financial interests.

