## [Peer review file · Nature Communications]

Reviewers' comments:

Reviewer #1 (Remarks to the Author):

Review of Düwel et al Nature hyperpolarized zymonic acid.

This manuscript describes a novel substrate for DNP hyperpolarization: zymonic acid (ZA); a cyclic butenolide natural product derivative of pyruvate. The chemical shift of two carbons: C1 and C5 are affected by pH in the physiological range and hence this is proposed as an indicator of tissue pH.

The pH of tissues is important and biomedically relevant, as it is altered in a number of pathological conditions, notably in solid tumors. There is evidence that it affects drug distribution, drug response and is an important contributor to the metastatic phenotype. Hence, there are compelling reasons to develop improved methods with which to measure pH in vivo.

There are a wide variety of MR, optical and PET/SPECT-based approaches with which to measure tissue pH. Of these, only APT has been used in humans, and it is of limited utility outside of the brain. Hence, the relevant questions are: (1) are there properties of ZA that are a significant improvement over the large number of other indicators available?; and (2) are there properties of ZA that make it more likely to be clinically translatable?

Regarding these questions, comparisons are made specifically the previously reported use of ^{13}C bicarbonate as a DNP pH indicator (Gallaher et al., Nature 2008).

Comments:

1. The solubility of ZA in DMSO is 4 M dissolved in 4 mL for a final concentration of 50 mM, which is sufficient to conduct DNP experiments in vivo. In comparison, the CsHCO_3 used by Gallaher is approx. 7M dissolved in 6 mL for a final concentration of 100mM
2. The pKa of 6.90 is optimal for an in vivo tracer. In comparison, the pKa of HCO_3 is ca. 6.2-6.3.
3. The DNP protocol resulted in 22% polarization, which is sufficient to conduct DNP experiments in vivo.
4. The relatively long T1s of 43 and 72s for C1 and C5 at 1 T are sufficient to conduct DNP experiments at that field. However, the T1s are shortened considerably at higher fields, (which are more common for in vivo imaging) to 17 and 16s for C1 and C5, respectively at 7T (by comparison, the T1 of HCO_3 is ca. 10 s). This occurs presumably via CSA but there could be some dipolar effects. Page 7, line 35, missing "...of ZA 'with deuterium would' reduce..."
5. ZA contains two titratable carbons, providing an internal reference (with a concomitant loss in dynamic range however). By comparison, the bicarbonate tracer is used in ratiometric mode, comparing HCO_3 and CO_2 (aq), with the latter being in vanishingly small quantities. Because bicarb is in slow exchange (on NMR timescale), newer methods have used magnetization transfer to measure with greater fidelity, yet are still challenging in a clinical setting. A greater dynamic range is available when comparing ZA to non-titratable urea, as a reference. In vivo, however, the internal reference is preferable as they are in the same

compartment.

6. The authors claim that ZA is non-toxic is substantiated by their supplementary data.

7. The relative chemical shifts of ZA do not appear to be significantly affected by calcium, temperature or concentration.

8. In vivo experiments included (a) injection of ZA/urea directly into bladder; (b) tail vein injection to normal rats (for kidney); and (c) tail vein injection in rats bearing a MAT B adenocarcinoma. The results in normal rat bladder and kidney were consistent with measurements (bladder) and literature (kidney). Notably, in the absence of a pH perturbation (which would be preferred) the investigators. Have measured intra- and extra-vascular pH in tumors, with the expected tumor pH being more acidic.

In conclusion, the current work proposes Zymonic acid (ZA) as a non-toxic and potentially clinically translatable pH indicator. The scientific and clinical question is very important and would be of general interest. There are potentially some important advantages of ZA over other DNP approaches to pH measurement. The methods were strong, and the characterization of this agent was relatively complete.

Reviewer #2 (Remarks to the Author):

This communication reports on the discovery of an interesting new hyperpolarized ^{13}C -labeled probe for measuring pH *in vivo*, ^{13}C -labeled zymonic acid. This probe has a number of nice features for measuring pH using hyperpolarized MR including; a long lifetime, only small chemical shift dependence on concentration, temperature, ionic strength and protein concentration. However, there are a number of concerns about the interpretation of the *in vivo* ZA data acquired as indicated in the specific comments below. Specifically, more convincing data needs to be provided about whether ZA is measuring an interstitial pH. While ZA represents an interesting new pH imaging candidate with characteristics favorable for clinical translation, it is clearly no closer to *in vivo* translation than several other previously published hyperpolarized ^{13}C MR approaches for measuring interstitial pH.

The following specific critiques are listed in order of appearance in the manuscript.

1. Abstract: The statement “However, currently no method to image extracellular pH has progressed beyond preclinical studies” should be removed from the abstract since it infers that ^{13}C -labeled zymonic acid has progressed beyond what has been done with other hyperpolarized probes such as ^{13}C bicarbonate.
2. Results, Synthesis of ZA and pH measurements *in vitro*, p. 2: It would be more physiologically relevant to report the solution-state T1 values in H₂O or buffer. Using a deuterated solvent can lengthen T1 values by reducing dipolar relaxation from the protons in water. This would also allow easier comparison with the apparent T1 values reported *in vivo*. Additionally, the T1 values should be reported without the addition of Gd chelates, as these may reduce the T1 value. In particular, the reported 50 μL formulated ZA with 5 mM Gd-chelate dissolved with 4 mL solvent would be expected to reduce the T1 values, by comparison with [1- ^{13}C] pyruvic acid. It would also be useful to know the field strength dependence of ZA (i.e., how much shorter is the T1 at the clinically relevant field strength of 3T).
3. Results, pH measurements *in vivo*, p. 3: T1s should be reported as “apparent T1”, since effects such as slice inflow/outflow and kidney filtration could affect these measurements. The apparent T1 measured for urea is also surprisingly short.
4. Results, Figure 4e and j: The fitting residuals should be provided along with the fits for all *in vivo* spectra to further substantiate all pH measurements. Based on the hyperpolarized spectra shown in Figure 4, there doesn't appear to be sufficient spectral resolution to justify fitting ZA in multiple tissue compartments in the kidney (cortex, medulla and urethral) or tissue and vascular compartments in the tumor.
5. Results: Judging from the reported kidney spectrum (figure 4e), more of the ZA signal arises from the cortex rather than the medulla. However, supplementary Figure 14 suggests that much more of the ZA ends up in the medulla rather than the cortex. These discrepant findings should be discussed.
6. Results: Figures 4g-i would benefit from a corresponding high-resolution anatomic imaging, more clearly showing the location of tumor and surrounding structures. Also, the color bar on figure 4i indicates a tumor pH which is above 7.2 which is pretty high for a solid tumor. The bar plot in figure 4k indicates that most of tumors studies had pH's less than 7, which is more consistent with what others have observed. A more representative example should be shown in figure 4 g-i.
7. Results: The authors state that “We detected an acidic tumor compartment 26 of pH 6.94 ± 0.12 ($n = 5$ rats, mean \pm s.d.) tentatively assigned to the extravascular compartment of the

tumor.” The manuscript would greatly benefit from more direct evidence that this probe is measuring the pH of the extravascular space of the tumor. For example direct comparison with another proven method of measuring interstitial pH, such as comparing it with a ³¹P-APP measure of interstitial pH (Gillies, R. J., Liu, Z. & Bhujwala, Z. ³¹P-MRS measurements of extracellular pH of tumors using 3-aminopropylphosphonate. The American journal of physiology 267, C195-203 (1994). This approach has been used to demonstrate the hyperpolarized ¹³C bicarbonate measures the interstitial pH of tumors.

8. Discussion: The author’s claim that ZA “allows an instantaneous assessment of pH whenever the agent reaches its target” is not supported by dynamic hyperpolarized ¹³C data demonstrating a constant pH over time after injection of ZA.

9. Discussion: The author’s claim that the real in-plane resolution of their technique is 1.31 mm is convoluted. The native in-plane resolution is 3.75 mm and zero-filling simply interpolates the data, rather than truly affecting the resolution? So when you take into account the impact of point spread function of the circular sampling scheme (a factor of 1.4), the in-plane resolution is increased to 5.25 mm not the reported 1.31 mm? This part of the discussion needs to be either re-written or removed

10. Discussion: The discussion of the evidence suggesting that ZA extravasates and predominantly stays in the extracellular space is confusing, mainly because of a lack of direct measurement of interstitial pH in vivo by other published means. Cells don't have a interstitial space, studies of the distribution of ZA in kidney tissues is complicated by the fact that kidneys are responsible for filtering ZA. I'm also not sure what the statements about the MCT transporters suggest about intracellular transport about ZA.

11. Supplementary Figure 8, p. 21: The authors should include histopathological images for exposed animals, not just unexposed.

Dear Reviewers,

Thank you for spending your time on reviewing our manuscript and for the constructive and helpful comments that you made on how to improve it.

We have addressed your comments both by additional *in vitro* and *in vivo* experiments and by adding and rephrasing parts of our manuscript.

Major experiments performed are first a comparison of our method with three other techniques that measure interstitial/extracellular pH in tumors (see new **Fig. 5e**), namely an invasive optical technique using a needle-type electrode as well as ^{31}P -MRS measurements using the pH probe 3-APP as suggested by referee #2 and *in vivo* and an *ex vivo* electrode pH measurements in excised tumor tissue. The pH values obtained from these methods agree well with the value determined for the extracellular pH compartment using ZA confirming that ZA can measure and distinguish extravascular (extracellular/interstitial) pH and intravascular pH.

Second, we have performed measurements of the longitudinal relaxation times of ZA at a 3 T clinical MRI scanner in a buffer based on H_2O as suggested by reviewer #2 (see new **Supporting Fig. 5**). With $T_1 = 43 \pm 3$ s (ZA_1 , $n = 3$) and $T_1 = 51 \pm 4$ s (ZA_5 , $n = 3$), we believe that the T_1 times of zymonic acid are long enough to successfully conduct *in vivo* experiments at the clinically relevant field strength of 3 T in the physiologically more relevant solution of 80 mM Tris buffer in H_2O .

Furthermore, we made clear that *in vivo* T_1 relaxation times reported throughout the manuscript have to be considered "apparent" T_1 times, we split the former Fig. 4 a-k into two figures **Fig. 4 a-f** (kidney) and **Fig. 5 a-e** (tumor) for better readability, we argued that a fast readout of the pH using ZA is possible based on the protonation/deprotonation effect which is fast on the NMR timescale with a speed of 10^3 - 10^4 Hz under physiological conditions, distinguished clearly between real and apparent in-plane resolution, clarified our statement about the lengthening of T_1 using deuteration, included histological images of animals exposed to ZA (**Supporting Fig. 8 e-h**), showed the fitting residuals and coefficients of determination from the kidney and tumor spectra (**Supplementary Fig. 14**), discussed the difference between the MRI and MALDI-MSI data of the distribution of ZA in the different kidney compartments (**Supplementary Fig. 15**) and included additional axial slices with proton images clearly depicting the location of the tumor and surrounding structures (**Supplementary Fig. 16**).

In our view, the manuscript has substantially improved from these changes. Again, we would like to express our gratitude to you for your input.

We hope that the points raised by you have been adequately addressed. We are looking forward to hearing from you and hope that we were able to convince you that the revised version of the manuscript will be acceptable for publication in *Nature Communications*.

Here, we would like to comment on your specific remarks:

Reviewer #1 (Remarks to the Author):

Review of Düwel et al Nature hyperpolarized zymonic acid.

This manuscript describes a novel substrate for DNP hyperpolarization: zymonic acid (ZA); a cyclic butenolide natural product derivative of pyruvate. The chemical shift of two carbons: C1 and C5 are affected by pH in the physiological range and hence this is proposed as an indicator of tissue pH.

The pH of tissues is important and biomedically relevant, as it is altered in a number of pathological conditions, notably in solid tumors. There is evidence that it affects drug distribution, drug response and is an important contributor to the metastatic phenotype. Hence, there are compelling reasons to develop improved methods with which to measure pH in vivo.

There are a wide variety of MR, optical and PET/SPECT-based approaches with which to measure tissue pH. Of these, only APT has been used in humans, and it is of limited utility outside of the brain. Hence, the relevant questions are: (1) are there properties of ZA that are a significant improvement over the large number of other indicators available?; and (2) are there properties of ZA that make it more likely to be clinically translatable?

Regarding these questions, comparisons are made specifically the previously reported use of ^{13}C bicarbonate as a DNP pH indicator (Gallaher et al., Nature 2008).

Comments:

- 1. The solubility of ZA in DMSO is 4 M dissolved in 4 mL for a final concentration of 50 mM, which is sufficient to conduct DNP experiments in vivo. In comparison, the CsHCO_3 used by Gallaher is approx. 7M dissolved in 6 mL for a final concentration of 100mM*

As both bicarbonate and zymonic acid could potentially act as a buffering substance and thus influence the pH of their surroundings, it is desirable to achieve a final concentration as low as possible while still being able to have a SNR large enough to be able to use the molecule for pH detection. As stated by the reviewer, cesium bicarbonate is usually being used at a final concentration of ~ 100 mM. Since high levels of cesium are toxic, it must be removed using an ion-exchange column. This requires additional time and thus results in a reduction of the polarization level at the time of injection. Gallagher et al. state that, although cesium bicarbonate has been used in animal experiments, this would not be practical in the clinic as even trace amounts of cesium could cause toxicity¹. When sodium bicarbonate is hyperpolarized directly, rather than indirectly using cesium bicarbonate, final concentrations of only 55 mM are reached². Another challenge of bicarbonate is the degassing of CO_2 after dissolution, further reducing the signal of bicarbonate available for acquisition.

Thus, it is advantageous that ZA can be used to detect pH with a sufficient SNR at a final concentration as low as 50 mM.

- 2. The pKa of 6.90 is optimal for an in vivo tracer. In comparison, the pKa of HCO_3 is ca. 6.2-6.3.*

We agree with the reviewer that the pKa of zymonic acid is optimal for an in vivo tracer and more convenient than the one of bicarbonate. Due to the $\text{pK}_a \approx 6.2-6.3$ of bicarbonate, more than one pH unit below the physiological pH ≈ 7.4 , the CO_2 signal is comparatively low and

noisy with respect to bicarbonate. Thus, only pH measurements at an acidic pH around the pK_a of bicarbonate give reliable results.

The $pK_a \approx 6.9$ of zymonic acid leads to a high sensitivity between pH 5.9 and pH 7.9, covering the physiologically relevant pH range.

3. *The DNP protocol resulted in 22% polarization, which is sufficient to conduct DNP experiments in vivo.*

The polarization level in the latest cesium bicarbonate publication was reported by Gallagher et al. to be 17 %³.

We thus agree that the polarization level of ZA is suitable to conduct DNP experiments, also supported by the *in vitro* and *in vivo* experiments shown in the paper.

4. *The relatively long T_1 s of 43 and 72s for C1 and C5 at 1 T are sufficient to conduct DNP experiments at that field. However, the T_1 s are shortened considerably at higher fields, (which are more common for in vivo imaging) to 17 and 16s for C1 and C5, respectively at 7T (by comparison, the T_1 of HCO_3 is ca. 10 s). This occurs presumably via CSA but there could be some dipolar effects. Page 7, line 35, missing "...of ZA 'with deuterium would' reduce..."*

We have corrected the sentence on Page 7, line 35, as suggested by the reviewer.

A common method to prolong the longitudinal relaxation time T_1 is deuteration, replacing protons with deuterium atoms. Replacing both the single proton attached to C_3 and the three protons from the methyl group C_6 lengthens T_1 . We are currently studying the effects of deuteration on ZA. Preliminary results suggest that at 1 T, T_1 of C_1 is lengthened by a factor of 1.3 and T_1 of C_5 is lengthened by a factor of 1.6 by deuteration, indicating that dipolar effects are indeed involved.

For hyperpolarized ^{13}C experiments, higher magnetic fields result in a larger chemical shift separation between the peaks of the substances of interest, which is an advantage, but higher magnetic fields also lead to shorter T_1 , which is a disadvantage. However, in contrast to conventional 1H experiments, higher fields do not result in higher signal. Thus, for hyperpolarized ^{13}C experiments, it would be best to perform the measurements at an intermediate and well shimmed magnetic field strength, e.g. at clinically widely available 3 T scanners, to balance the chemical shift separation and T_1 .

We have thus measured the solution-state T_1 value of hyperpolarized zymonic acid in 80 mM Tris buffer in H_2O at the clinically available field strength of 3 T without the addition of Dotarem as suggested by reviewer #2 in comment #2. We report the longitudinal relaxation times (T_1) at 3 T in 80 mM Tris buffer in H_2O on page 2 of the manuscript, we have adjusted the appropriate sentences in the methods section "Hyperpolarization with dissolution DNP" and have replaced **Supplementary Fig. 5** accordingly:

"This resulted in a solution polarization level for ZA of 22 ± 2 % ($n = 3$, see Methods) with relatively long longitudinal relaxation times (T_1) at 3 T for both ZA_1 ($T_1 = 43 \pm 3$ s, $n = 3$) and ZA_5 ($T_1 = 51 \pm 4$ s, $n = 3$) in 80 mM Tris buffer in H_2O (**Supplementary Fig. 5**)."

" T_1 measurements of natural abundance ZA in 80 mM Tris (Sigma Aldrich, USA) buffer in H_2O were performed on a clinical Biograph mMR MR-PET (Siemens, Germany, $B_0 = 3$ T) using a pulse length of 0.2 ms, a flip angle of 15 degrees, a spectral bandwidth of 2500 Hz and a repetition time of 5 s. Carbon center frequency and flip angle were determined using an 8 M ^{13}C -urea phantom with 5 mM Dotarem (Guerbet, France) and 0.1% sodium azide (Sigma-Aldrich, USA)."

Supplementary Figure 5 | Longitudinal relaxation time T_1 of hyperpolarized natural abundance ZA *in vitro* at 3 T. A three-parameter monoexponential curve was fitted to each dataset and the mean and standard deviation was calculated from the resulting decay constants of 50 mM ZA in 80 mM Tris buffer in H_2O adjusted with 1M NaOH to an average pH of 6.53 ± 0.03 at 27 °C. The close proximity of the frequently and fast exchanging proton of the hydroxy group attached to carbon number two of ZA most likely causes the shorter T_1 of carbon number one ($^{13}ZA_1$, **a**) compared to carbon number five ($^{13}ZA_5$, **b**) of ZA *in vitro*.

In conclusion, we believe that the longitudinal relaxation times T_1 of zymonic acid are large enough even at higher field strengths in order to successfully conduct *in vivo* experiments at the clinically relevant field strength of 3 T. In addition, T_1 of zymonic acid at an even higher field strength of 7 T is longer than the respective T_1 of bicarbonate, as already stated by the reviewer, and well in line with the T_1 of the well-established agent urea, reported in our answer to reviewer #2, comment #3.

5. ZA contains two titratable carbons, providing an internal reference (with a concomitant loss in dynamic range however). By comparison, the bicarbonate tracer is used in ratiometric mode, comparing HCO_3^- and $CO_2(aq)$, with the latter being in vanishingly small quantities. Because bicarb is in slow exchange (on NMR timescale), newer methods have used magnetization transfer to measure with greater fidelity, yet are still challenging in a clinical setting. A greater dynamic range is available when comparing ZA to non-titratable urea, as a reference. *In vivo*, however, the internal reference is preferable as they are in the same compartment.

^{13}C magnetization transfer measurements with hyperpolarized bicarbonate have been used by Gallagher et al.^{3,4} to determine the rate constant describing flux between bicarbonate and $CO_2(aq)$ caused by chemical exchange between these two molecules, in order to study the activity of the enzyme carbonic anhydrase. To our knowledge, these magnetization transfer measurements cannot be used to determine pH. However, in the same publication, the pH was determined from the HCO_3^-/CO_2 ratio as usual. From these experiments, Gallagher et al.³ have concluded that the pH determined using the HCO_3^-/CO_2 ratio can be overestimated if equilibration of the ^{13}C labels is slow on the NMR timescale.

Supplementary Fig. 17 compares the pH values derived from the different dynamic ranges available from the chemical shifts of the two zymonic acid peaks compared to urea (greater dynamic range, higher accuracy) and from the chemical shifts of the two zymonic acid peaks compared to each other (smaller dynamic range, lower accuracy).

Supplementary Figure 17 | ^{13}C biosensor pH of the same buffer phantom measurement evaluated from ZA with and without considering the additional urea peak used as chemical shift reference at 7 T. The ^{13}C biosensor pH was back-calculated based on the chemical shift difference of both ^{13}C -labeled ZA positions with respect to the pH insensitive ^{13}C urea (a) and based on the chemical shift difference between the two ^{13}C -labeled ZA positions only (b). c, The pH values extracted from the two ^{13}C pH maps correlate well with the electrode pH (in white in a and b). At the limit of its sensitivity (at $\text{pH} \approx 5$), the back-calculation of the ^{13}C biosensor pH is improved by taking the urea peak into account as pH insensitive chemical shift reference.

In all of the spectra acquired *in vivo* (Fig. 3d, Fig. 4e, Fig. 5d), we only observe a single urea peak, which can thus be used to determine B_0 effects independent of urea being in intra- or extracellular and independent of ZA being intra- or extracellular. Breukels et al.⁵ detected a 3-Hz chemical shift difference between intracellular and extracellular lactate. If one would observe a difference in chemical shift on the same order of magnitude for ZA and/or urea as for lactate, this change would have a very small effect on the pH determined from this method of 0.01 pH units.

In conclusion, we believe that one can either use the external reference with the advantage of a greater dynamic range and higher accuracy but with the drawback of having to copolarize and inject two substances simultaneously or one can use the internal reference with the drawback of a smaller dynamic range and lower accuracy but with the advantage of having to polarize and inject zymonic acid only.

6. *The authors claim that ZA is non-toxic is substantiated by their supplementary data.*

In addition to the non-toxicity data already shown in **Supplementary Fig. 6** (cytotoxicity tests show that ZA is non-toxic within experimentally relevant concentration ranges), **Supplementary Fig. 7** (dose escalation study testing for *in vivo* toxicity of ZA in three rats), **Supplementary Fig. 9** (non-toxicity of ZA was substantiated by blood collection for hematology and clinical chemistry before and 24 hours (acute), 7, 21 and 30 days (subacute) after injection of ZA) and **Supplementary Table 1** (toxicopathological study) we have included histopathological images for animals exposed to ZA in **Supplementary Fig. 8** (a toxicopathological study shows non-ZA-associated alterations both within exposed and unexposed animals) in response to reviewer #2, comment #11:

Supplementary Figure 8 | A toxicopathological study shows non-ZA-associated alterations both within exposed and unexposed animals. Representative images of background alterations observed histopathologically in liver (**a**, **e**), intestines (**b**, **f**), kidney (**c**, **g**) and pancreas (**d**, **h**) in animals after NaCl administration (upper row) and after fivefold overdosage of ZA (lower row). **a**, **e**, In the liver, slight periportal infiltration predominantly with lymphocytes was observed in the periportal region (arrows) and intralobular (arrowheads). **b**, **f**, Slight mixed infiltration and fibrosis of the villi occurred in all parts of the intestines. **c**, **g**, In the kidney, intraepithelial (arrows) and intraluminal (arrowheads) hyaline droplets within the proximal tubuli were observed only in male rats regardless of the injected compound. **d**, One of the control animals showed a focal acinar-to-ductular metaplasia within the pancreas. The arrows indicate metaplastic ductular formations. **h**, Normal pancreatic tissue in an animal after fivefold overdosage of ZA. (H&E staining, bars 50 μm).

7. *The relative chemical shifts of ZA do not appear to be significantly affected by calcium, temperature or concentration.*

In addition to the experiments supporting our claim mentioned by the reviewer that the relative chemical shifts of ZA do not appear to be significantly affected by calcium, temperature or concentration (**Fig. 2c**, **Supplementary Fig. 10**), we show data that the pH measured using ZA is unaffected by ionic strength or protein concentration within the physiological ranges:

In vitro experiments find the slope of the acid dissociation constant (ΔpKa) as a function of ionic strength (ΔI) to be $\Delta\text{pKa}/\Delta\text{I} = -0.7 \times 10^{-3}$ pH/mM and thus show that the pH uncertainty in the physiological range with an ionic strength of 135 to 165 mM results in ≈ 0.02 pH units (included as **Supplementary Fig. 11**).

Further *in vitro* experiments find the slope of the chemical shift (Δcs) as a function of protein concentration (Δpc) to be $\Delta\text{cs}/\Delta\text{pc} = -0.7 \times 10^{-3}$ $\Delta\text{ppm}/(\text{g/L})$ and thus show that the pH uncertainty in the physiological range with a protein concentration of 60 to 80 g/L results in ≈ 0.01 pH units (included as **Supplementary Fig. 12**).

Relaxivities do not affect the pH determination of the method at hand since we do not use a ratiometric method.

8. *In vivo* experiments included (a) injection of ZA/urea directly into bladder; (b) tail vein injection to normal rats (for kidney); and (c) tail vein injection in rats bearing a MAT B adenocarcinoma. The results in normal rat bladder and kidney were consistent with measurements (bladder) and literature (kidney). Notably, in the absence of a pH perturbation (which would be preferred) the investigators have measured intra- and extra-vascular pH in tumors, with the expected tumor pH being more acidic.

In conclusion, the current work proposes Zymonic acid (ZA) as a non-toxic and potentially clinically translatable pH indicator. The scientific and clinical question is very important and would be of general interest. There are potentially some important advantages of ZA over other DNP approaches to pH measurement. The methods were strong, and the characterization of this agent was relatively complete.

We agree with the reviewer, that the scientific and clinical question of measuring pH non-invasively is very important and of general interest and that ZA potentially possesses some important advantages over other DNP approaches to pH measurement, as demonstrated by our *in vivo* measurements in bladder (**Fig. 3**), kidney (**Fig. 4**) and tumor (**Fig. 5**).

Reviewer #2 (Remarks to the Author):

This communication reports on the discovery of an interesting new hyperpolarized ¹³C-labeled probe for measuring pH in vivo, ¹³C-labeled zymonic acid. This probe has number of nice features for measuring pH using hyperpolarized MR including; a long lifetime, only small chemical shift dependence on concentration, temperature, ionic strength and protein concentration. However, there are number of concerns about the interpretation of the in vivo ZA data acquired as indicated in the specific comments below. Specifically, more convincing data needs to be provided about whether ZA is measuring an interstitial pH. While ZA represents an interesting new pH imaging candidate with characteristics favorable for clinical translation, it is clearly no closer to in vivo translation than several other previously published hyperpolarized ¹³C MR approaches for measuring interstitial pH.

The following specific critiques are listed in order of appearance in the manuscript.

1. *Abstract: The statement "However, currently no method to image extracellular pH has progressed beyond preclinical studies" should be removed from the abstract since it infers that ¹³C-labeled zymonic acid has progressed beyond what has been done with other hyperpolarized probes such as ¹³C bicarbonate.*

As suggested by the reviewer, we have removed the statement that no extracellular pH imaging method has progressed beyond preclinical studies from the abstract.

2. *Results, Synthesis of ZA and pH measurements in vitro, p. 2: It would be more physiologically relevant to report the solution-state T₁ values in H₂O or buffer. Using a deuterated solvent can lengthen T₁ values by reducing dipolar relaxation from the protons in water. This would also allow easier comparison with the apparent T₁ values reported in vivo. Additionally, the T₁ values should be reported without the addition of Gd chelates, as these may reduce the T₁ value. In particular, the reported 50 μ L formulated ZA with 5 mM Gd-chelate dissolved with 4 mL solvent would be expected to reduce the T₁ values, by comparison with [¹³C] pyruvic acid. It would also be useful to know the field strength dependence of ZA (i.e., how much shorter is the T₁ at the clinically relevant field strength of 3T).*

Although we do not expect a large effect of the Gd chelate on the T₁ of ZA due to the low final concentration of $\approx 63 \mu$ M of Dotarem after dissolution, we agree that it is of high interest to know the T₁ of ZA at the clinically relevant field strength of 3 T. We have thus

followed the reviewer’s suggestion to measure the solution-state T_1 value of zymonic acid in 80 mM Tris buffer in H_2O at 3 T after hyperpolarization without the addition of Dotarem. We changed the reported parameters on page 2 of the manuscript, have adjusted the appropriate sentences in the methods section “Hyperpolarization with dissolution DNP” and have replaced **Supplementary Fig. 5** accordingly (see also our response to reviewer #1, comment #4):

“This resulted in a solution polarization level for ZA of $22 \pm 2 \%$ ($n = 3$, see Methods) with relatively long longitudinal relaxation times (T_1) at 3 T for both ZA_1 ($T_1 = 43 \pm 3$ s, $n = 3$) and ZA_5 ($T_1 = 51 \pm 4$ s, $n = 3$) in 80 mM Tris buffer in H_2O (**Supplementary Fig. 5**).”

“ T_1 measurements of natural abundance ZA in 80 mM Tris (Sigma Aldrich, USA) buffer in H_2O were performed on a clinical Biograph mMR MR-PET (Siemens, Germany, $B_0 = 3$ T) using a pulse length of 0.2 ms, a flip angle of 15 degrees, a spectral bandwidth of 2500 Hz and a repetition time of 5 s. Carbon center frequency and flip angle were determined using an 8 M ^{13}C -urea phantom with 5 mM Dotarem (Guerbet, France) and 0.1% sodium azide (Sigma-Aldrich, USA).”

Supplementary Figure 5 | Longitudinal relaxation time T_1 of hyperpolarized natural abundance ZA *in vitro* at 3 T. A three-parameter monoexponential curve was fitted to each dataset and the mean and standard deviation was calculated from the resulting decay constants of 50 mM ZA in 80 mM Tris buffer in H_2O adjusted with 1M NaOH to an average pH of 6.53 ± 0.03 at 27 °C. The close proximity of the frequently and fast exchanging proton of the hydroxy group attached to carbon number two of ZA most likely causes the shorter T_1 of carbon number one ($^{13}ZA_1$, **a**) compared to carbon number five ($^{13}ZA_5$, **b**) of ZA *in vitro*.

In conclusion, we believe that the longitudinal relaxation times T_1 of zymonic acid are large enough to successfully conduct *in vivo* experiments at the clinically relevant field strength of 3 T in the physiologically more relevant solution of 80 mM Tris buffer in H_2O .

3. Results, pH measurements *in vivo*, p. 3: T_1 s should be reported as as “apparent T_1 ”, since effects such as slice inflow/outflow and kidney filtration could affect these measurements. The apparent T_1 measured for urea is also surprisingly short.

We agree with the reviewer that slice inflow/outflow and excitation profile effects affect the measurement and the reported T_1 values. We therefore follow the reviewer’s suggestion and state all T_1 values that were measured *in vivo* as “apparent T_1 ” values in the revised manuscript.

The surprisingly short apparent T_1 measured for urea may be attributed to the high field strength of 7 T at which the experiments were performed. Golman⁶ reports an *in vivo* T_1 of 20

± 2 s for urea at 2.35 T, von Morze⁷ measured an *in vivo* T_1 of 15 s for urea at 3 T and Bahrami⁸ reports an even shorter *in vivo* T_1 of 7-13 s for urea at 3 T. As the T_1 of urea is expected to decrease with increasing field strength⁷, our reported *in vivo* apparent T_1 of 13 ± 2 s for urea at 7 T can be attributed to the comparatively higher field strength at which our *in vivo* measurements are performed.

4. *Results, Figure 4e and j: The fitting residuals should be provided along with the fits for all in vivo spectra to further substantiate all pH measurements. Based on the hyperpolarized spectra shown in Figure 4, there doesn't appear to be sufficient spectral resolution to justify fitting ZA in multiple tissue compartments in the kidney (cortex, medulla and urethral) or tissue and vascular compartments in the tumor.*

We have followed the reviewer's suggestion and are providing the fitting residuals for all fits of *in vivo* spectra to further substantiate the pH measurements (**Supplementary Fig. 14**):

Supplementary Figure 14 | Representative fits and fitting residuals for multiple tissue compartments in the kidneys and in the tumor. a-c, In the kidneys, increasing the number of fitted zymonic acid peak pairs from one (a, $R^2 = 0.77$) to two (b, $R^2 = 0.92$) to three (c, $R^2 = 0.95$) results in a reduction of the fitting residuals (red line) and an improved coefficient of determination R^2 . d-e, Analogously, increasing the number of fitted zymonic acid peaks pairs from one (e, $R^2 = 0.83$) to two (f, $R^2 = 0.88$) results in a reduction of the fitting residuals (red line) and an improved coefficient of determination R^2 in the tumor. Urea (0 ppm) and parapyruvate hydrate (15.7 ppm) are fitted in all spectra.

Based on the fitting residuals shown in red above and in **Supplementary Fig. 14** and based on the increase of the coefficient of determination R^2 both in the kidney and in the tumor when adjusting the number of fitted ZA peak pairs to the expected number of compartments, we believe that the shown spectra provide sufficient spectral resolution to justify fitting ZA in

multiple tissue compartments in the kidney (**Fig. 4e**) as well as tissue and vascular compartments in the tumor (**Fig. 5d**, formerly Fig. 4j).

5. *Results: Judging from the reported kidney spectrum (figure 4e), more of the ZA signal arises from the cortex rather than the medulla. However, supplementary Figure 14 suggests that much more of the ZA ends up in the medulla rather than the cortex. These discrepant findings should be discussed.*

This apparent discrepancy can be explained by the different times at which the distribution of ZA is measured by MRI compared to MALDI (ZA is measured already 10 s after injection and MALDI distribution of ZA is fixed only after 2-3 minutes, where one would expect more ZA to be involved in the renal filtering process in the tissue area containing the medulla and calyx).

We have added an explanation of this finding to the figure legend of **Supplementary Fig. 15**:

“MALDI-MSI represents the distribution of ZA within the kidney fixed 2-3 minutes after injection whereas hyperpolarized MRI shows the distribution of ZA within the kidney 10 s after injection. Whereas in the hyperpolarized MR image, shortly after injection, the cortex exhibits the largest contribution to the overall signal, in MALDI-MSI, much longer after injection, more ZA is already involved in the renal filtering process and thus the area containing the medulla and calyx show the largest signal contribution.”

6. *Results: Figures 4g-i would benefit from a corresponding high-resolution anatomic imaging, more clearly showing the location of tumor and surrounding structures. Also, the color bar on figure 4i indicates a tumor pH which is above 7.2 which is pretty high for a solid tumor. The bar plot in figure 4k indicates that most of tumors studies had pH's less than 7, which is more consistent with what others have observed. A more representative example should nbe shown in figure 4 g-i.*

As suggested, we have included additional axial slices of the tumor shown in **Fig. 5a-c** (formerly Fig. 4 g-i) in a new **Supplementary Fig. 16** clearly depicting the location of the tumor and surrounding structures.

Supplementary Figure 16 | Axial slices from the animal shown in Fig. 5 bearing a Mat B III tumor (arrow). a-p, Proton images with a field of view of 6 cm were acquired every 1 mm using a fast spin echo sequence (see Methods). **g-k**, The five proton images contained within the 5 mm thick hyperpolarized ^{13}C image are marked with a blue box. Image **(i)** represents the central tumor slice and coincides with the center of the 5 mm thick hyperpolarized ^{13}C image.

The color bar on **Fig. 5c** (formerly Fig. 4i), in analogy to **Fig. 4c**, shows the mean biosensor pH, which has contributions from both intravascular and extravascular (interstitial/extracellular) pH. Therefore, the displayed mean pH value of a pH around 7.2 is higher than the pH in the interstitial compartment alone, which is at pH 7.10.

7. *Results: The authors state that “We detected an acidic tumor compartment 26 of $\text{pH } 6.94 \pm 0.12$ ($n = 5$ rats, mean \pm s.d.) tentatively assigned to the extravascular compartment of the tumor.” The manuscript would greatly benefit from more direct evidence that this probe is measuring the pH of the extravascular space of the tumor. For example direct comparison with another proven method of measuring interstitial pH, such as comparing it with a ^{31}P -APP measure of interstitial pH (Gillies, R. J., Liu, Z. & Bhujwala, Z. ^{31}P -MRS measurements of extracellular pH of tumors using 3-aminopropylphosphonate. The American journal of physiology 267, C195-203 (1994). This approach has been used to demonstrate the hyperpolarized ^{13}C bicarbonate measures the interstitial pH of tumors.*

We agree with the reviewer that our manuscript was lacking direct evidence that our probe is

measuring the pH of the extravascular space of the tumor. We have therefore performed additional animal experiments providing a comparison of our method with three other techniques that measure interstitial/extracellular pH in tumors, namely an invasive optical technique using a needle electrode as well as ^{31}P -MRS measurements using the pH probe 3-APP as suggested by referee #2 and *in vivo* and an *ex vivo* electrode pH measurements in excised tumor tissue. The pH values obtained from these methods agree well with the value determined for the extravascular (extracellular/interstitial) pH compartment using ZA, confirming that ZA measures predominantly extravascular (extracellular/interstitial) pH and intravascular pH.

We have included an additional section in the manuscript and have included a new panel in **Fig. 5** (formerly Fig. 4 g-k):

“We detected an acidic tumor compartment of $\text{pH } 6.94 \pm 0.12$ ($n = 5$ rats, mean \pm s.d.) tentatively assigned to the extravascular (interstitial/extracellular) compartment of the tumor. The signal stemming from the other tumor compartment of $\text{pH } 7.40 \pm 0.05$ ($n = 5$ rats, mean \pm s.d.) was tentatively assigned to the vasculature of the tumor. Furthermore, we observed a pH of 7.39 ± 0.05 ($n = 5$ rats, mean \pm s.d.) in blood near the vena cava (**Fig. 5c-e**). These measured tumor pH values were compared to three other independent interstitial/extracellular pH measurement methods in MAT B III tumor bearing rats ($n = 4$ rats, mean \pm s.d.): First, the extracellular ^{31}P pH-probe 3-aminopropylphosphonate (3-APP)^{23,24} showed an extracellular pH of 7.11 ± 0.04 *in vivo*; second, the pH determined using a needle-type optical sensor was 6.84 ± 0.13 *in vivo*; third, *ex vivo* measurement of tumor tissue using a standard pH microelectrode gave a pH of 6.82 ± 0.12 (**Fig. 5e**). These three pH values agree well with the value determined for the extravascular (extracellular/interstitial) pH compartment using ZA (no significant difference, $p > 0.05$, Mann-Whitney-Wilcoxon test) suggesting that ZA measures and distinguishes extravascular (extracellular/interstitial) pH and intravascular pH.”

Figure 5e | Hyperpolarized ZA *in vivo* pH measurements show an acidic tumor pH at 7 T. [...] Spectra from tumor voxels (**d**, shown for one representative animal) are best fitted with two pairs of ZA peaks (red, blue) and consistently show different pH values in the tumor compared to the vena cava, demonstrated in five animals (**e**, individual datapoints and mean \pm s.d.) and compared with three other interstitial/extracellular pH measurements in four animals, namely ^{31}P MRS *in vivo* using 3-APP, a needle-type optical sensor *in vivo* and an *ex vivo* tissue electrode pH measurement.

8. *Discussion: The author's claim that ZA "allows an instantaneous assessment of pH whenever the agent reaches its target" is not supported by dynamic hyperpolarized ^{13}C data demonstrating a constant pH over time after injection of ZA.*

The pH-measurement based on pH-dependent ^{13}C -resonances of ZA only relies on the deprotonation/protonation of its OH group at $\text{pK}_a = 6.90$. Proton exchange of OH groups takes place on the order of 10^3 - 10^4 Hz under physiological conditions⁹. Equilibration of ZA according to the actual pH can therefore be assumed to occur instantaneously on the timescale of an NMR experiment which is on the order of seconds. Importantly, no additional enzymes are involved in this equilibration process. Even if reduced diffusion would slow down proton exchange, the equilibration of ZA's protonated and deprotonated forms would still be fast compared to the timescale of the NMR experiment.

We have rephrased our statement in the revised manuscript by including the relevant timescale of ZA's pH mechanism instead of the term "instantaneous":

"Importantly, ZA's pH sensitivity arises from protonation and deprotonation in proximity to its ^{13}C -labeled sites, which is fast on the NMR timescale (10^3 - 10^4 Hz under physiological conditions⁹) and allows an assessment of pH whenever the agent reaches its target."

9. *Discussion: The author's claim that the real in-plane resolution of their technique is 1.31 mm is convoluted. The native in-plane resolution is 3.75 mm and zero-filling simply interpolates the data, rather than truly affecting the resolution? So when you take into account the impact of point spread function of the circular sampling scheme (a factor of 1.4), the in-plane resolution is increased to 5.25 mm not the reported 1.31mm? This part of the discussion needs to be either re-written or removed*

As suggested by the referee we have more clearly distinguished the nominal, real and apparent resolution in the revised manuscript:

"The FIDCSI sequence used in this work (see Methods) has a slice thickness of 5 mm and a nominal in-plane resolution of 3.75 mm. Zero-filling by a factor of four results in an apparent in-plane resolution of 0.94 mm. Taking into account the shape of the point spread function of the circular sampling scheme, the in-plane resolution is affected by a factor of ≈ 1.40 , resulting in a real in plane-resolution of 5.25 mm and an apparent in-plane resolution of 1.31 mm."

10. *Discussion: The discussion of the evidence suggesting that ZA extravasates and predominantly stays in the extracellular space is confusing, mainly because of a lack of direct measurement of interstitial pH in vivo by other published means. Cells don't have a interstitial space, studies of the distribution of ZA in kidney tissues is complicated by the fact that kidneys are responsible for filtering ZA. I'm also not sure what the statements about the MCT transporters suggest about intracellular transport about ZA.*

As discussed in our response to reviewer #2, comment #11, we have performed additional animal experiments providing a comparison of our method with three other techniques that measure interstitial/extracellular pH in tumors and have shown that the ZA tumor pH measurements of the extravascular (extracellular/interstitial) pH compartment agree well with those methods. We have therefore rewritten and expanded our discussion in the manuscript:

"Evidence suggests that ZA extravasates and predominantly stays in the extracellular/interstitial space, thus measuring extracellular pH. Extravasation of ZA to renal tissue is confirmed by MALDI-MSI in kidney slices (**Supplementary Fig. 15**). Comparison of

the pH measured using ZA in MAT B III tumor bearing rats with the pH values determined by three other interstitial/extracellular pH measurements (^{31}P MRS, needle-type optical sensor, electrode) showed good agreement between these methods, suggesting that ZA can measure and distinguish both extravascular (interstitial/extracellular) and intravascular pH (**Fig. 5e**)."

The cell experiments, even though cells do not have interstitial space, suggest that ZA does not enter the cytosol and therefore does not measure intracellular pH which is in line with the *in vivo* experiments. We have therefore kept the cell experiment in the manuscript but have removed the rather vague statements regarding MCT transport as suggested by the reviewer.

11. Supplementary Figure 8, p. 21: The authors should include histopathological images for exposed animals, not just unexposed.

As suggested by the reviewer, we added histopathological images for animals exposed to ZA to **Supplementary Fig. 8**.

Supplementary Figure 8 | A toxicopathological study shows non-ZA-associated alterations both within exposed and unexposed animals. Representative images of background alterations observed histopathologically in liver (**a, e**), intestines (**b, f**), kidney (**c, g**) and pancreas (**d, h**) in animals after NaCl administration (upper row) and after fivefold overdosage of ZA (lower row). **a, e**, In the liver, slight periportal infiltration predominantly with lymphocytes was observed in the periportal region (arrows) and intralobular (arrowheads). **b, f**, Slight mixed infiltration and fibrosis of the villi occurred in all parts of the intestines. **c, g**, In the kidney, intraepithelial (arrows) and intraluminal (arrowheads) hyaline droplets within the proximal tubuli were observed only in male rats regardless of the injected compound. **d**, One of the control animals showed a focal acinar-to-ductular metaplasia within the pancreas. The arrows indicate metaplastic ductular formations. **h**, Normal pancreatic tissue in an animal after fivefold overdosage of ZA. (H&E staining, bars 50 μm).

References

- 1 Gallagher, F. A., Kettunen, M. I. & Brindle, K. M. Imaging pH with hyperpolarized ¹³C. *NMR Biomed.* **24**, 1006-1015, doi:10.1002/nbm.1742 (2011).
- 2 Wilson, D. M. *et al.* Multi-compound polarization by DNP allows simultaneous assessment of multiple enzymatic activities in vivo. *J. Magn. Reson.* **205**, 141-147, doi:10.1016/j.jmr.2010.04.012 (2010).
- 3 Gallagher, F. A. *et al.* Carbonic Anhydrase Activity Monitored In Vivo by Hyperpolarized ¹³C-Magnetic Resonance Spectroscopy Demonstrates Its Importance for pH Regulation in Tumors. *Cancer Res.* **75**, 4109-4118, doi:10.1158/0008-5472.CAN-15-0857 (2015).
- 4 Gallagher, F. A. *et al.* Magnetic resonance imaging of pH in vivo using hyperpolarized ¹³C-labelled bicarbonate. *Nature* **453**, 940-943, doi:10.1038/nature07017 (2008).
- 5 Breukels, V. *et al.* Direct dynamic measurement of intracellular and extracellular lactate in small-volume cell suspensions with (¹³C) hyperpolarised NMR. *NMR Biomed.* **28**, 1040-1048, doi:10.1002/nbm.3341 (2015).
- 6 Golman, K., Ardenkjaer-Larsen, J. H., Petersson, J. S., Mansson, S. & Leunbach, I. Molecular imaging with endogenous substances. *Proc. Natl. Acad. Sci. U. S. A.* **100**, 10435-10439, doi:10.1073/pnas.1733836100 (2003).
- 7 von Morze, C. *et al.* Imaging of Blood Flow Using Hyperpolarized [¹³C] Urea in Preclinical Cancer Models. *J. Magn. Reson. Imaging* **33**, 692-697, doi:10.1002/jmri.22484 (2011).
- 8 Bahrami, N., Swisher, C. L., Von Morze, C., Vigneron, D. B. & Larson, P. E. Kinetic and perfusion modeling of hyperpolarized (¹³C) pyruvate and urea in cancer with arbitrary RF flip angles. *Quant Imaging Med Surg* **4**, 24-32, doi:10.3978/j.issn.2223-4292.2014.02.02 (2014).
- 9 Liepinsh, E. & Otting, G. Proton exchange rates from amino acid side chains - Implications for image contrast. *Magn. Reson. Med.* **35**, 30-42, doi:DOI 10.1002/mrm.1910350106 (1996).
- 10 Ackerman, J. J. H. & Neil, J. J. in *Diffusion MRI* (ed D. Jones) 110-124 (Oxford University Press, 2011).

REVIEWERS' COMMENTS:

Reviewer #1 (Remarks to the Author):

This remains a very strong contribution that has been improved by addressing reviewer 2's comments. ZA does, indeed appear to have significant strengths compared to bicarbonate. One minor correction in the introduction, APT is not exclusively intra-cellular. Data from van Zijl supports that the majority of amides being interrogated are extracellular. since the source of MT effects is not known with certainty, I should suggest removing the word "intracellular".

Reviewer #2 (Remarks to the Author):

The authors have done a thorough job of responding to the prior critiques of this manuscript and the manuscript has been substantially improved. While there remains concern that there isn't sufficient spectral resolution to robustly fitting ZA in multiple tissue compartments or tissue/vascular compartments, the findings of this well done study is worthy of publication.

Dear Reviewers,

Thank you for spending your time on reviewing our manuscript and for the constructive and helpful comments that you made on how to improve it.

In our view, the manuscript has substantially improved during the review process. Again, we would like to express our gratitude to you for your input.

We hope that the points raised by you have been adequately addressed and that the revised version of the manuscript will be acceptable for publication in *Nature Communications*.

Here, we would like to comment on your specific remarks:

Reviewer #1 (Remarks to the Author):

This remains a very strong contribution that has been improved by addressing reviewer 2's comments. ZA does, indeed appear to have significant strengths compared to bicarbonate. One minor correction in the introduction, APT is not exclusively intra-cellular. Data from van Zijl supports that the majority of amides being interrogated are extracellular. since the source of MT effects is not known with certainty, I should suggest removing the word "intracellular".

We agree with the reviewer that the pH measured using APT CEST is not exclusively intracellular. However, to the best of our knowledge, the APT MRI signal stems predominantly from intracellular proteins and peptides, as reported by van Zijl and other groups¹⁻³. For tissues, in which intra- and extracellular pH quickly equilibrate, as suggested in the case of an ischemic stroke, it was reasoned that the APT CEST signal would also reflect tissue pH^{4,5}. For other tissues, such as tumors, which maintain a pH difference between intra- and extracellular compartments⁶, APT CEST is reported to measure predominantly intracellular pH⁷. Therefore, the compartmental pH measured by endogenous APT CEST differs from the one measured using exogenous pH agents such as 3-APP, bicarbonate or CEST agents, which mainly interrogate extracellular pH.

As suggested by the reviewer we avoided stating that APT CEST measures exclusively intracellular pH and we clarified that the amide proton signals only predominantly stem from intracellular proteins:

„Endogenous amide proton transfer (APT) chemical exchange saturation transfer (CEST) experiments⁸ utilizing the pH-dependent proton exchange from predominantly intracellular proteins are currently used to study pH in the brain *in human*.“

Reviewer #2 (Remarks to the Author):

The authors have done a thorough job of responding to the prior critiques of this manuscript and the manuscript has been substantially improved. While there remains concern that there isn't sufficient spectral resolution to robustly fitting ZA in multiple tissue compartments or tissue/vascular compartments, the findings of this well done study is worthy of publication.

Indeed, improving the spectral resolution of ZA's pH-dependent peaks would largely benefit the fitting of ZA-spectra from tissue/vascular compartments. For cases, where a good separation of the compartments would not be possible, we could alternatively envision to

calculate pH profile parameters, such as mean, weighted median, mode(s) and skewness from ZA's peak distribution as demonstrated previously using 3-APP as a pH-reporter⁸.

We have added the following sentence to the discussion on page 5:

“Although our fitting residuals shown in Supplementary Fig. 15 show that fitting ZA in multiple tissue compartments in kidney and tumor is reasonable, complete separation and fitting of extravascular and intravascular pH compartments remains challenging for the case of small pH differences between individual compartments.”

References

- 1 Zhou, J. Y., Payen, J. F., Wilson, D. A., Traystman, R. J. & van Zijl, P. C. M. Using the amide proton signals of intracellular proteins and peptides to detect pH effects in MRI. *Nat. Med.* **9**, 1085-1090, doi:10.1038/nm907 (2003).
- 2 Wen, Z. *et al.* MR imaging of high-grade brain tumors using endogenous protein and peptide-based contrast. *Neuroimage* **51**, 616-622, doi:10.1016/j.neuroimage.2010.02.050 (2010).
- 3 van Zijl, P. C. & Yadav, N. N. Chemical exchange saturation transfer (CEST): what is in a name and what isn't? *Magn. Reson. Med.* **65**, 927-948, doi:10.1002/mrm.22761 (2011).
- 4 Sun, P. Z., Cheung, J. S., Wang, E. & Lo, E. H. Association between pH-weighted endogenous amide proton chemical exchange saturation transfer MRI and tissue lactic acidosis during acute ischemic stroke. *J. Cereb. Blood Flow Metab.* **31**, 1743-1750, doi:10.1038/jcbfm.2011.23 (2011).
- 5 Sun, P. Z., Wang, E. & Cheung, J. S. Imaging acute ischemic tissue acidosis with pH-sensitive endogenous amide proton transfer (APT) MRI--correction of tissue relaxation and concomitant RF irradiation effects toward mapping quantitative cerebral tissue pH. *Neuroimage* **60**, 1-6, doi:10.1016/j.neuroimage.2011.11.091 (2012).
- 6 Vaupel, P., Kallinowski, F. & Okunieff, P. Blood flow, oxygen and nutrient supply, and metabolic microenvironment of human tumors: a review. *Cancer Res.* **49**, 6449-6465 (1989).
- 7 McVicar, N., Li, A. X., Meakin, S. O. & Bartha, R. Imaging chemical exchange saturation transfer (CEST) effects following tumor-selective acidification using Iodine. *NMR Biomed.* **28**, 566-575, doi:10.1002/nbm.3287 (2015).
- 8 Lutz, N. W., Le Fur, Y., Chiche, J., Pouyssegur, J. & Cozzone, P. J. Quantitative in vivo characterization of intracellular and extracellular pH profiles in heterogeneous tumors: a novel method enabling multiparametric pH analysis. *Cancer Res.* **73**, 4616-4628, doi:10.1158/0008-5472.CAN-13-0767 (2013).